# Pairwise Worst-Case Ratio Analysis for Discriminative Dimensionality Reduction via Minorization-Maximization

## Abstract

In this paper, we investigate a novel discriminative dimensionality reduction method based on maximizing the minimum pairwise ratio of between-class to within-class scatter. This objective function enhances class separability by providing critical, adaptive control over the variance within each class pair. The resulting max-min fractional programming problem is non-convex and notoriously challenging to solve. Our key contribution is a provably convergent, two-level iterative algorithm, termed GDMM-QF (generalized Dinkelbach-minorization-maximization for quadratic fractional programs), to find a high-quality solution. The outer loop employs a generalized Dinkelbach-type procedure to transform the fractional program into an equivalent sequence of subtractive-form max-min subproblems. For the inner loop, we develop an efficient minorization-maximization (MM) algorithm that tackles the non-convex subproblem by iteratively solving a simple quadratic program (QP), which we derive from the dual of a convex surrogate. The proposed GDMM-QF framework is computationally efficient, guaranteed to converge, and requires no hyperparameter tuning. Experiments on multiple benchmark datasets confirm the superiority of our method in learning discriminative projections, consistently achieving lower classification error than state-of-the-art alternatives.

## 1 Introduction

Modern data acquisition technologies have led to the proliferation of high-dimensional datasets, where the number of measured features can be exceptionally large. This phenomenon, often termed the "curse of dimensionality," introduces significant challenges, including increased computational overhead, a heightened risk of model overfitting due to spurious correlations, and a considerable loss of data interpretability. To address these issues in the context of supervised classification, dimensionality reduction has become an indispensable preprocessing step. Methodologies for dimensionality reduction are broadly categorized into two main paradigms: feature extraction, which constructs a new, smaller set of features by projecting the data onto a lower-dimensional manifold Nie et al. (2021b); Wang et al. (2024); Nie et al. (2023; 2021a); Chang et al. (2016); Nie et al. (2017); Li et al. (2018a), and feature selection, which aims to identify and retain only the most informative subset of the original features Gui et al. (2017); Li et al. (2017); Sheikhpour et al. (2017); Hancer et al. (2020); Li et al. (2022); Shen et al. (2021); Li et al. (2018b); Luo et al. (2018). Both approaches seek to produce a more compact and meaningful data representation, thereby enhancing classification accuracy, preventing overfitting, and improving the comprehensibility of the resulting model.

The core objective of discriminant analysis is to identify a linear projection that optimally separates distinct classes within a dataset. This family of supervised methods has evolved significantly since the seminal work of Fisher, with researchers developing a wide array of class separability metrics over the decades Fisher (1936); Rao (1948); Bian & Tao (2011b); Zhang & Yeung (2010); Yu et al. (2011); Su et al. (2015); Nie et al. (2021b); Wang et al. (2024). Among these, linear discriminant analysis (LDA) remains the most prominent. Originally conceived by Fisher for binary classification Fisher (1936) and later extended by Rao to handle multiple classes Rao (1948), LDA seeks a subspace projection that maximizes the ratio of between-class variance to within-class variance.

Despite its widespread use, classical LDA exhibits several critical limitations. It is susceptible to performance degradation when sample sizes are small, which can yield unreliable covariance matrix estimates and violate its underlying Gaussian assumptions Nie et al. (2020a;b). It is also known to be sensitive to outliers Nie et al. (2021b). A particularly significant drawback arises in multi-class scenarios where the target dimension is less than the number of classes minus one. In such cases, the LDA objective, which effectively averages pairwise class separations, becomes biased by the most distant class pair, causing less separated pairs to be projected even closer, potentially leading to class overlap Hamsici & Martinez (2008). This phenomenon is known as the "worst-case class separation" problem.

Several algorithms improve multiclass discrimination by adaptively weighting class distances, but they often falter on the closest pair of classes. More recently, researchers have reframed the problem in a worst-case light Omati et al. (2025); Song et al. (2017); Xu et al. (2010; 2012); Shao & Sang (2014); Ding et al. (2014); Shao & Sang (2017); Li et al. (2015); Hu et al. (2014); Zhang & Yeung (2010); Bian & Tao (2011a). For example, Bian et al. Bian & Tao (2011a) identify the pair of classes with the smallest distance between their means—the "worst-case" pair that nearly overlaps—and introduce a max–min distance analysis (MMDA) to enlarge that minimal gap in the reduced subspace. Zhang et al. build upon the MMDA with worst-case LDA (WLDA) to further enhance separation under challenging scenarios. Extensions in Su et al. (2015) and Omati et al. (2025) move MMDA into heteroscedastic settings, using the Chernoff distance to maximize the smallest inter-class divergence while controlling the intra-class variance. It is worth mentioning that Bian & Tao (2011b); Su et al. (2015) pioneered the conversion of high-dimensional datasets into moderate-dimensional representations by applying principal component analysis (PCA) as a preprocessing step in their proposed algorithms. This approach enables the algorithms to operate effectively in substantially lower dimensions while preserving most of the energy of the data, depending on the reduction coefficient. For example, a dataset with 1024 dimensions can be reduced to fewer than 50 dimensions while retaining 98% of the energy Bian & Tao (2011b); Su et al. (2015); Omati et al. (2025); Wang et al. (2024).

While these advanced methods improve upon classical LDA, strategies that focus solely on maximizing the minimum distance between class centroids (whether Euclidean or Chernoff-based) are incomplete. They fail to account for the internal dispersion, or within-class scatter, of each class. Consequently, even if the means of two classes are pushed apart, the classes themselves may still overlap if characterized by high variance. To address this interplay between separation and compactness, the worst-case ratio analysis (WCRA) objective was proposed, which seeks to maximize the minimum ratio of between-class to within-class scatter.

A notable attempt to solve this problem was made by Wang et al. Wang et al. (2024). Their approach reformulates the non-convex fractional program by iteratively transforming it into a sequence of quadratic subproblems. They then apply a semidefinite relaxation to arrive at a tractable semidefinite program (SDP). However, this method has two key drawbacks: the relaxation is not guaranteed to be tight, potentially leading to suboptimal solutions, and its performance depends on the tuning of at least two hyperparameters.

Crucially, a fundamental limitation of the approach in Wang et al. (2024) is its reliance on a global within-class scatter matrix in the denominator of the ratio. This non-pairwise normalization fails to adapt to the specific compactness of the most challenging class pairs, limiting its effectiveness in scenarios where class variances differ significantly.

In this paper, we introduce a novel and provably convergent iterative algorithm, termed GDMM-QF (generalized Dinkelbach-minorization-maximization for quadratic fractional programs), for solving the pairwise worst-case ratio analysis (PWCRA) problem. Our approach is designed to solve a more adaptive version of the worst-case separation problem. Instead of normalizing by a global measure of compactness, our objective function evaluates the separability of each class pair relative to its own unique within-class scatter. The effectiveness of this formulation becomes more pronounced when class variances differ significantly.

To solve the PWCRA problem, our GDMM-QF algorithm employs a nested iterative process. For the outer loop, we adapt a generalized Dinkelbach-type procedure to transform the challenging max-min fractional objective into an equivalent max-min subtractive problem. We provide a rigorous proof that this outer iterative framework, which can be viewed as a specialized Newton's method,

is guaranteed to converge to the global optimum of the ratio problem. The inner subproblem at each Dinkelbach iteration, however, remains a non-convex max-min program. To solve this, we employ the minorization-maximization (MM) principle for the inner loop. We construct a tight, convex surrogate that lower-bounds the true objective, resulting in a semidefinite program (SDP) at each inner step. Crucially, we show that this SDP can be solved even more rapidly by formulating its dual, which takes the form of a simple quadratic program (QP). This dual-loop structure with proven convergence at both levels makes GDMM-QF a robust and computationally efficient overall algorithm that is also fully parameter-free.

The remainder of this paper is structured as follows. In Section 2, we formulate the PWCRA problem. Section 3 presents our proposed two-level optimization algorithm, which uses a generalized Dinkelbach procedure for the outer loop and the MM approach to solve the inner loop's subproblem. We then evaluate its performance through extensive experiments in Section 4. Finally, Section 5 provides concluding remarks.

## 2 PROBLEM FORMULATION

Consider a dataset $\mathbf{X} = [\mathbf{x}_1, \dots, \mathbf{x}_n] \in \mathbb{R}^{d \times n}$ containing $n$ samples distributed across $C$ distinct classes. The objective is to learn a linear transformation matrix $\mathbf{T} \in \mathbb{R}^{d \times m}$ (where $m \ll d$) that projects the high-dimensional data into a lower-dimensional subspace. The transformation is designed to simultaneously maximize between-class separation and minimize within-class dispersion.

To formalize this, we define class-pairwise scatter matrices. For any pair of classes $(i, j)$, the between-class scatter $\mathbf{S}_b^{ij}$ and within-class scatter $\mathbf{S}_w^{ij}$ are given by:

$$\mathbf{S}_b^{ij} = (\bar{\mathbf{x}}_i - \bar{\mathbf{x}}_j)(\bar{\mathbf{x}}_i - \bar{\mathbf{x}}_j)^T, \qquad \mathbf{S}_w^{ij} = \sum_{h \in \{i,j\}} \sum_{\mathbf{x}_k \in \tau_h} (\mathbf{x}_k - \bar{\mathbf{x}}_h)(\mathbf{x}_k - \bar{\mathbf{x}}_h)^T, \qquad (1)$$

where $\tau_k$ denotes the set of samples belonging to the $k$-th class and $\bar{\mathbf{x}}_k$ is the corresponding class mean. Using these definitions, the PWCRA problem is formulated as the following optimization:

$$\max_{\mathbf{T}^T \mathbf{T} = \mathbf{I}_m} \min_{1 \le i < j \le C} \frac{\text{tr}\left(\mathbf{T}^T \mathbf{S}_b^{ij} \mathbf{T}\right)}{\text{tr}\left(\mathbf{T}^T \mathbf{S}_w^{ij} \mathbf{T}\right)}. \qquad (2)$$

The core of this formulation is the max-min objective, which guarantees pairwise class separability by maximizing the minimum performance ratio across all class pairs. This directly addresses the "worst-case" separation problem. Moreover, the use of pairwise scatter matrices allows the model to adaptively handle the unique covariance structure inherent to each class pair, a significant advantage over methods that rely on a single, global within-class scatter matrix. The orthogonality constraint $\mathbf{T}^T \mathbf{T} = \mathbf{I}_m$ is imposed to ensure the projected features are uncorrelated and to provide a unique basis for the solution subspace.

The following sections detail our approach to solving the challenging non-convex problem in (2).

## 3 A TWO-LEVEL OPTIMIZATION ALGORITHM FOR PWCRA

The PWCRA optimization problem in (2) is a non-convex, max-min fractional program, which is inherently difficult to solve directly. To this end, we propose a provably convergent, two-level iterative algorithm. The outer level employs a generalized Dinkelbach-type procedure to transform the fractional objective into a more manageable subtractive form. The inner level then solves the resulting non-convex max-min subproblem using a specialized algorithm derived from the Minorization-Maximization (MM) framework.

### 3.1 OUTER LOOP: GENERALIZED DINKELBACH PROCEDURE FOR FRACTIONAL PROGRAMMING

The PWCRA problem belongs to the class of general max-min ratio problems:

$$\max_{\mathbf{T} \in \mathcal{X}} \min_{1 \le i < j \le C} \frac{f_{ij}(\mathbf{T})}{g_{ij}(\mathbf{T})}, \qquad (3)$$

where $f_{ij}(\mathbf{T}) = \text{tr}(\mathbf{T}^T \mathbf{S}_b^{ij} \mathbf{T})$, $g_{ij}(\mathbf{T}) = \text{tr}(\mathbf{T}^T \mathbf{S}_w^{ij} \mathbf{T}) > 0$, and the feasible set is the Stiefel manifold $\mathcal{X} = \{\mathbf{T} \in \mathbb{R}^{d \times m} \mid \mathbf{T}^T \mathbf{T} = \mathbf{I}_m\}$. To solve this max-min fractional program, we generalize Dinkelbach's algorithm Dinkelbach (1967). The theoretical foundation of this iterative approach is captured in the following theorems.

**Theorem 1.** *The global optimum of the general max-min ratio problem $\max_{\mathbf{T} \in \mathcal{X}} \min_{ij} \frac{f_{ij}(\mathbf{T})}{g_{ij}(\mathbf{T})}$ is equivalent to finding the largest root $\lambda^*$ of the function $h(\lambda) = \max_{\mathbf{T} \in \mathcal{X}} \min_{ij} \{f_{ij}(\mathbf{T}) - \lambda g_{ij}(\mathbf{T})\}$.*

*Proof.* See Appendix B. $\square$

**Theorem 2.** *The iterative procedure outlined in Algorithm 1 is equivalent to Newton's method applied to find the root of the function $h(\lambda)$.*

*Proof.* See Appendix C. $\square$

A summary of the generalized Dinkelbach procedure is outlined in Algorithm 1, which can be found in the Appendix G. Given the iterate $\mathbf{T}^k$, we compute the current worst-case ratio $\lambda_k$. This $\lambda_k$ is then used to form a subtractive max-min subproblem (line 4), the solution of which becomes the next iterate $\mathbf{T}^{k+1}$.

The convergence properties of this algorithm are stated below, guaranteeing its desirable behavior.

**Theorem 3.** *Algorithm 1 monotonically increases the objective value of the PWCRA problem at each iteration and converges to the global optimal solution.*

*Proof.* See Appendix D. $\square$

### 3.2 INNER LOOP: SOLVING THE MAX-MIN SUBPROBLEM VIA MM

Each outer loop iteration requires solving a problem of the form:

$$\max_{\mathbf{T}^T \mathbf{T} = \mathbf{I}_m} \min_{1 \le i < j \le C} \text{tr}\left(\mathbf{T}^T \tilde{\mathbf{S}}_{Cij} \mathbf{T}\right), \tag{4}$$

where $\tilde{\mathbf{S}}_{Cij} \triangleq \mathbf{S}_b^{ij} - \lambda_k \mathbf{S}_w^{ij}$. To facilitate the development of our proposed algorithm for solving the problem in (4), we first introduce and prove a key lemma.

**Lemma 1.** *Given that the matrices $\tilde{\mathbf{S}}_{Cij}$ are positive semi-definite, the original non-convex semi-orthogonality constraint, $\mathbf{T}^T \mathbf{T} = \mathbf{I}$, as stated in problem (4), can be substituted with the less strict condition $\mathbf{T}^T \mathbf{T} \preccurlyeq \mathbf{I}$. This relaxation is valid because the global maximizer of the resulting relaxed problem will inherently satisfy the original equality constraint.*

*Proof.* The proof is provided in Appendix E. $\square$

Leveraging Lemma 1, we can now reformulate the initial problem from (4) into the following relaxed optimization problem:

$$\max_{\mathbf{T}^T \mathbf{T} \preccurlyeq \mathbf{I}_m} \min_{1 \le i < j \le C} \text{tr}\left(\mathbf{T}^T \tilde{\mathbf{S}}_{Cij} \mathbf{T}\right). \tag{5}$$

A significant advantage of this reformulation is that the new constraint in (5) is convex. This is because the inequality $\mathbf{T}^T \mathbf{T} \preccurlyeq \mathbf{I}_m$ is equivalent to the linear matrix inequality (LMI) $\begin{bmatrix} \mathbf{I}_m & \mathbf{T}^T \\ \mathbf{T} & \mathbf{I}_d \end{bmatrix} \succcurlyeq 0$.

However, the optimization in (5) is a challenging non-convex problem due to its max-min structure, where each term $h_{ij}(\mathbf{T}) = \text{tr}(\mathbf{T}^T \tilde{\mathbf{S}}_{Cij} \mathbf{T})$ is convex in $\mathbf{T}$ (note that the potential indefiniteness of the matrices $\tilde{\mathbf{S}}_{ij}$ is not a concern, as each term can be convexified by shifting $\tilde{\mathbf{S}}_{ij}$ with a suitable scalar matrix $\alpha \mathbf{I}$—for example, using $\alpha \ge \max_{i,j} \lambda_{\max}(-\tilde{\mathbf{S}}_{ij})$—an operation that does not alter the optimizer due to the constraint $\mathbf{T}^T \mathbf{T} = \mathbf{I}_m$). To address (5), we employ the minorization-maximization for max-min (MM4MM) approach Saini et al. (2024), an overview of which is detailed in Appendix A.

At each iteration $t$, we construct a tractable surrogate function by minorizing each of the convex quadratic terms, $h_{ij}(\mathbf{T})$. We replace each term with its first-order Taylor expansion around the current estimate $\mathbf{T}^t$. This tangent hyperplane serves as a tight lower bound:

$$h_{ij}(\mathbf{T}) = \operatorname{tr}\left(\mathbf{T}^T \tilde{\mathbf{S}}_{Cij}\mathbf{T}\right) \geq \operatorname{tr}\left(\left(\mathbf{T}^t\right)^T \tilde{\mathbf{S}}_{Cij}\mathbf{T}^t\right) + 2\operatorname{tr}\left(\left(\mathbf{T}^t\right)^T \tilde{\mathbf{S}}_{Cij}\left(\mathbf{T} - \mathbf{T}^t\right)\right)$$

$$= 2\operatorname{tr}\left(\left(\mathbf{T}^t\right)^T \tilde{\mathbf{S}}_{Cij}\mathbf{T}\right) - \operatorname{tr}\left(\left(\mathbf{T}^t\right)^T \tilde{\mathbf{S}}_{Cij}\mathbf{T}^t\right) \triangleq \tilde{h}_{ij}(\mathbf{T}). \quad (6)$$

Substituting these linear lower bounds $\tilde{h}_{ij}(\mathbf{T})$ back into the original problem (5) yields the surrogate problem for the MM update. This problem involves maximizing the minimum of these linear functions:

$$\max_{\mathbf{T}^T\mathbf{T}\preccurlyeq\mathbf{I}_m} \min_{1\leq i<j\leq C} \quad 2\operatorname{tr}\left(\mathbf{A}_{ij}^T\mathbf{T}\right) + c_{ij}, \quad (7)$$

where

$$\mathbf{A}_{ij}^T \triangleq \left(\mathbf{T}^t\right)^T \tilde{\mathbf{S}}_{Cij}, \quad (8)$$

$$c_{ij} = -\operatorname{tr}\left(\left(\mathbf{T}^t\right)^T \tilde{\mathbf{S}}_{Cij}\mathbf{T}^t\right), \quad (9)$$

which are constants within the current iteration. Although this surrogate problem (7) is convex and can be solved as an SDP, we can devise a more efficient solution method.

To develop this more efficient solver, we begin by reformulating the inner minimization over the discrete indices $(i, j)$. This is accomplished by introducing a set of continuous auxiliary variables, $\{z_{ij}\}$, which are constrained to the probability simplex ($\sum z_{ij} = 1, z_{ij} \geq 0$). This transformation recasts the original problem into the equivalent max-min formulation shown in (10):

$$\max_{\mathbf{T}^T\mathbf{T}\preccurlyeq\mathbf{I}_m} \min_{\{z_{ij}\}} \quad 2\operatorname{tr}\left(\mathbf{A}(\mathbf{z})^T\mathbf{T}\right) + \sum_{1\leq i<j\leq C} z_{ij}c_{ij} \quad \text{s.t.} \quad z_{ij} \geq 0, \sum_{1\leq i<j\leq C} z_{ij} = 1, \quad (10)$$

where the matrix $\mathbf{A}$ is now defined as the convex combination $\mathbf{A}(\mathbf{z}) \triangleq \sum_{1\leq i<j\leq C} z_{ij}\mathbf{A}_{ij}$.

Let the objective function in (10) be $L(\mathbf{T}, \mathbf{z})$. This function is affine (and thus concave) with respect to the maximization variable $\mathbf{T}$ and affine (and thus convex) with respect to the minimization variable $\mathbf{z}$. Given that the optimization domains for both $\mathbf{T}$ and $\mathbf{z}$ are compact and convex, the conditions of Sion's minimax theorem Sion (1958) are met. This allows interchanging the operators, leading to the following equivalent formulation:

$$\min_{\{z_{ij}\}} \max_{\mathbf{T}^T\mathbf{T}\preccurlyeq\mathbf{I}_m} \quad 2\operatorname{tr}\left(\mathbf{A}(\mathbf{z})^T\mathbf{T}\right) + \sum_{1\leq i<j\leq C} z_{ij}c_{ij} \quad \text{s.t.} \quad z_{ij} \geq 0, \sum_{1\leq i<j\leq C} z_{ij} = 1. \quad (11)$$

This dual formulation is highly advantageous, as the inner maximization problem in (11) now admits a closed-form analytical solution. Focusing on the trace term in the objective, the Von Neumann inequality Marshall (1979) states that $\operatorname{tr}\left(\mathbf{A}^T\mathbf{T}\right) \leq \sum_{k=1}^{m}\sigma_k(\mathbf{A})$, where $\sigma_k(\cdot)$ denotes the $k$-th singular value. This upper bound is attained when $\mathbf{T}$ is set to $\mathbf{T}^* = \mathbf{A}\left(\mathbf{A}^T\mathbf{A}\right)^{-\frac{1}{2}}$. Critically, this optimal $\mathbf{T}^*$ inherently fulfills the strict semi-orthogonality constraint $\left(\mathbf{T}^*\right)^T\mathbf{T}^* = \mathbf{I}$. By substituting this analytical solution for $\mathbf{T}^*$ into the dual problem (11), we eliminate the variable $\mathbf{T}$ and arrive at the following optimization problem solely over $\mathbf{z}$:

$$\min_{\{z_{ij}\}} \quad 2\sum_{i=1}^{m}\sigma_i(\mathbf{A}(\mathbf{z})) + \sum_{1\leq i<j\leq C} z_{ij}c_{ij} \quad \text{s.t.} \quad z_{ij} \geq 0, \sum_{1\leq i<j\leq C} z_{ij} = 1, \quad (12)$$

where the dependency of $\mathbf{A}$ on $\mathbf{z}$ is made explicit. The first term in this objective is exactly twice the nuclear norm of $\mathbf{A}(\mathbf{z})$, denoted $\|\mathbf{A}(\mathbf{z})\|_*$. Since the nuclear norm is a convex function, and $\mathbf{A}(\mathbf{z})$ is a linear function of $\mathbf{z}$, problem (12) is convex. While it can be reformulated and solved as an SDP Recht et al. (2010), it offers a computational advantage over (7) due to having fewer variables and constraints.

Once the optimal $\mathbf{z}^*$ is found by solving (12), the corresponding update for $\mathbf{T}$ is computed as:

$$\mathbf{T}^{(t+1)} = \mathbf{A}\left(\mathbf{z}^*\right)\left(\mathbf{A}^T\left(\mathbf{z}^*\right)\mathbf{A}\left(\mathbf{z}^*\right)\right)^{-\frac{1}{2}}. \quad (13)$$

This $\mathbf{T}^{(t+1)}$ becomes the input for the subsequent iteration, and the process repeats until convergence as outlined in Algorithm 2 (see Appendix G).

The primary computational bottleneck within each iteration of this approach is solving the convex optimization problem (12). As noted previously, this problem can be reformulated and solved as a SDP. However, standard interior-point methods for SDPs have a high computational complexity, scaling polynomially with the matrix dimensions. For this problem, the cost is approximately $\mathcal{O}((d+m)^{4.5})$ per iteration, which can be prohibitive for large-scale datasets. This high cost motivates the development of a more efficient method for solving the subproblem (12). To reduce this computational burden, we now introduce an alternative approach based on the MM principle, applied directly to the challenging nuclear norm term in (12). This strategy replaces the expensive SDP with a sequence of much simpler QPs.

We begin by rewriting problem (12) to explicitly show the nuclear norm and the linear term in summation form:

$$\min_{\{z_{ij}\}} \quad 2\|\mathbf{A}(\mathbf{z})\|_* + \sum_{1 \leq i < j \leq C} z_{ij} c_{ij} \text{s.t.} \qquad z_{ij} \geq 0, \qquad \sum_{1 \leq i < j \leq C} z_{ij} = 1. \tag{14}$$

The core idea is to replace the non-smooth nuclear norm $\|\mathbf{A}(\mathbf{z})\|_*$ with a smooth, quadratic upper bound at each iteration. To this end, we employ a variational form of the nuclear norm, which expresses it as a joint minimization problem:

$$\|\mathbf{X}\|_* = \frac{1}{2} \min_{\boldsymbol{\Phi} \succ \mathbf{0}} \operatorname{tr}(\mathbf{X}^T \mathbf{X} \boldsymbol{\Phi}) + \operatorname{tr}(\boldsymbol{\Phi}^{-1}). \tag{15}$$

Using (15), we can reformulate problem (14) into an equivalent joint minimization problem over both $\mathbf{z}$ and an auxiliary positive definite matrix $\boldsymbol{\Phi}$:

$$\min_{\{z_{ij}\}, \boldsymbol{\Phi} \succ \mathbf{0}} \quad \operatorname{tr}\left(\mathbf{A}(\mathbf{z})^T \mathbf{A}(\mathbf{z}) \boldsymbol{\Phi}\right) + \operatorname{tr}(\boldsymbol{\Phi}^{-1}) + \sum_{1 \leq i < j \leq C} z_{ij} c_{ij} \text{s.t.} \qquad z_{ij} \geq 0, \qquad \sum_{1 \leq i < j \leq C} z_{ij} = 1. \tag{16}$$

Problem (16) can be tackled using an alternating minimization method. In this scheme, for a given $\mathbf{z}$ at iterate $t$, denoted $\mathbf{z}^t$, we first minimize (16) with respect to $\boldsymbol{\Phi}$. The problem has a closed-form solution for the optimal $\boldsymbol{\Phi}$, given by $\boldsymbol{\Phi}^* = (\mathbf{A}(\mathbf{z}^k)^T \mathbf{A}(\mathbf{z}^k))^{-\frac{1}{2}}$ . Substituting $\boldsymbol{\Phi}$ back into the objective yields the following minimization problem over $\mathbf{z}$:

$$\min_{\{z_{ij}\}} \quad \operatorname{tr}\left(\mathbf{A}(\mathbf{z})^T \mathbf{A}(\mathbf{z}) \boldsymbol{\Phi}^k\right) + \sum_{1 \leq i < j \leq C} z_{ij} c_{ij} \quad \text{s.t.} \quad z_{ij} \geq 0, \qquad \sum_{1 \leq i < j \leq C} z_{ij} = 1. \tag{17}$$

This problem can be transformed into a standard QP of the form:

$$\min_{\{z_{ij}\}} \quad \mathbf{z}^T \mathbf{Q} \mathbf{z} + \sum_{1 \leq i < j \leq C} z_{ij} c_{ij} \quad \text{s.t.} \quad z_{ij} \geq 0, \sum_{1 \leq i < j \leq C} z_{ij} = 1, \tag{18}$$

where the matrix $\mathbf{Q}$ is constructed as follows:

$$\mathbf{Q} = \mathbf{S}^{\mathrm{T}}(\mathbf{I} \otimes \boldsymbol{\Phi}^t)\mathbf{S}. \tag{19}$$

Here, the symbol $\otimes$ represents the Kronecker product and $\mathbf{S}$ is formed by stacking the vectorized matrices $\mathbf{A}_{ij}$. Specifically, $\mathbf{S} = [\boldsymbol{v}_1, \boldsymbol{v}_2, \ldots, \boldsymbol{v}_K]$ with $K = C(C-1)/2$, and each column $\boldsymbol{v}_k = \vec{(\mathbf{A}_{ij})}$ corresponds to a unique pair $(i, j)$ via the index mapping $k = \frac{(i-1)(2C-i)}{2} + j - i/$ The detailed derivation of $\mathbf{Q}$ is available in Appendix F.

This QP formulation (18) is computationally advantageous. It can be solved with standard solvers, and the combined cost of updating $\boldsymbol{\Phi}^t$ and solving the QP is approximately $\mathcal{O}(C^6 + d^3)$, which is substantially lower than the $\mathcal{O}((d+m)^{4.5})$ complexity of the original SDP approach in (12). While this QP needs to be resolved iteratively for updated $\boldsymbol{\Phi}^t$ matrices, the process converges rapidly, typically in under 10 iterations Omati et al. (2025). The full procedure for using this alternating minimization to find the optimal $\mathbf{z}^*$ is detailed in Algorithm 3 (see Appendix G).

## 4 NUMERICAL RESULTS

### 4.1 DATASETS

We evaluate the performance of the our proposed algorithm on four real-world datasets from the UCI repository. These datasets are briefly described below:

The **COIL-20 dataset** contains 1,440 grayscale images of 20 different objects, with 72 images per object taken at pose intervals of 5 degrees. Each image is $32 \times 32$ pixels, resulting in 1,024-dimensional feature vectors. This dataset presents challenges in object recognition under varying viewpoints and is commonly used for evaluating dimensionality reduction techniques in computer vision applications. The **Diamond dataset** comprises 599 instances with four main features: carat weight, depth, table size, and clarity. The labels represent the quality of the cut, categorized into four classes: Fair, Good, Ideal, and Premium. This dataset tests the algorithm's ability to handle regression-like data with continuous features and ordinal class relationships. The **Yale Face Database** consists of grayscale face images of 15 individuals, with 11 images per person captured under different lighting conditions and facial expressions. Each image is $32 \times 32$ pixels (1,024 dimensions after vectorization). This dataset is particularly challenging due to significant variations in illumination and expression while maintaining the same identity, making it ideal for testing discriminative dimensionality reduction methods. The **Iris dataset** consists of 150 instances, each represented by four features: sepal length, sepal width, petal length, and petal width. The label categorizes each instance into one of three classes: Iris-setosa, Iris-versicolor, or Iris-virginica, with 50 samples per class. We use the Iris dataset solely for visualization purposes. This choice is a direct consequence of our evaluation protocol, adapted from Su et al. (2015); Omati et al. (2025), which considers comparing methods at their individual optimal projection dimensions. For high-dimensional datasets, these optimal dimensions vary widely between algorithms. Consequently, for the purpose of 2D visualization, projecting them all to a fixed 2D space would be unfair. In contrast, the Iris dataset's inherent low dimensionality makes a 2D projection a suitable and fair ground for visually comparing the class separability achieved by all methods.

## 4.2 Experimental Protocol and Compared Methods

To refer to our approach in the experimental results, we name our proposed method GDMM-QF (PWCRA), reflecting its design to solve the PWCRA problem—a max-min quadratic-fractional program—using a generalized Dinkelbach and minorization-maximization framework.

For comparison, we included several widely used discriminant analysis methods: linear discriminant analysis (LDA) Fisher (1936); Rao (1948), max-min Distance analysis (MMDA) Bian & Tao (2011b), weighted heteroscedastic max-min distance analysis (WHMMDA) Su et al. (2018; 2015), $\ell_{1,2}$ LDA Nie et al. (2021b), MM4MM (QP-MMDA) Omati et al. (2025), and max-min ratio analysis (MMRA) Wang et al. (2024).

We randomly split each high-dimensional dataset in half—50% of the samples for training and the remaining 50% for testing. As a preprocessing step, following the protocol of Omati et al. (2025); Wang et al. (2024); Su et al. (2015), we apply PCA to project all feature vectors down to 50 dimensions, thereby retaining over 98% of the total variance. We repeat this entire process 20 times with independent random splits and report the mean accuracy and its standard deviation.

A key aspect of our evaluation is how the target dimensionality is selected. Because each method—including ours and the baselines—reaches its peak accuracy at a distinct target dimensionality, as mentioned before, we follow the evaluation protocol from Su et al. (2015); Omati et al. (2025) and report results at each method's respective optimal dimensionality. To find this optimal value, the original dimensionality $d$ was reduced to various potential values from 1 to $d-1$ for each method. The only exception was for LDA, where the maximum dimensionality of the selected subspace was constrained to $C-1$ to achieve its best performance and allow for a fair comparison.

Classification in the reduced subspaces was performed using three classifiers: the nearest neighbor classifier (1-NN), the nearest mean classifier (NM), and the quadratic discriminant analysis (QDA). The quadratic classifier utilized the following decision rule:

$$\mathbf{x} \in \arg \min_{i=1,\ldots,C} \left\{ (\mathbf{x} - \bar{\mathbf{x}}_i)^T \, \boldsymbol{\Sigma}_i^{-1} \, (\mathbf{x} - \bar{\mathbf{x}}_i) \; + \; \log|\boldsymbol{\Sigma}_i| \right\},$$

where $\bar{\mathbf{x}}_i$ represents the mean vector of class $i$, and $\boldsymbol{\Sigma}_i$ is the covariance matrix of class $i$. This choice of classifiers ensured that the methods could be evaluated on their ability to create both linearly and non-linearly separable subspaces, providing a thorough assessment of performance.

All experiments were conducted in MATLAB R2022b on a dual-socket Intel Xeon E5-2695 v3 workstation equipped with $2 \times 14$ cores (56 threads total), operating at a base frequency of 2.3 GHz (up to 3.3 GHz turbo) and featuring 70 MiB of L3 cache.

Table 1: Results for COIL-20, Diamond and Yale

| COIL-20 Dataset | | | |
|---|---|---|---|
| Method | Classifier | | |
| | 1-NN | NM | QDA |
| LDA Fisher (1936); Rao (1948) | 0.0046 (16, Std:0.0036) | 0.0340 (16, Std:0.0070) | 0.0130 (11, Std:0.0062) |
| MMDA Bian & Tao (2011b) | 0.0119 (16, Std:0.0054) | 0.0360 (36, Std:0.0062) | 0.0128 (16, Std:0.0054) |
| WHMMDA Su et al. (2018; 2015) | 0.0167 (16, Std:0.0053) | 0.0388 (46, Std:0.0086) | 0.0191 (31, Std:0.0074) |
| $\ell_{1,2}$ LDA Nie et al. (2021b) | 0.0059 (38, Std:0.0021) | 0.0360 (46, Std:0.0084) | 0.0071 (11, Std:0.0037) |
| MM4MM (QP-MMDA) Omati et al. (2025) | 0.0134 (16, Std:0.0058) | 0.0383 (46, Std:0.0087) | 0.0158 (16, Std:0.0052) |
| MMRA Wang et al. (2024) | 0.0051 (41, Std:0.0031) | 0.0358 (41, Std:0.0072) | 0.0243 (6, Std:0.0096) |
| GDMM-QF (PWCRA) | **0.0026** (26, Std:0.0014) | **0.0312** (36, Std:0.0055) | **0.0109** (16, Std:0.0031) |
| Diamond Dataset | | | |
| Method | Classifier | | |
| | 1-NN | NM | QDA |
| LDA Fisher (1936); Rao (1948) | 0.0484 (2, Std:0.0162) | 0.0835 (3, Std:0.0405) | 0.0318 (3, Std:0.0217) |
| MMDA Bian & Tao (2011b) | 0.0484 (2, Std:0.0170) | 0.0852 (3, Std:0.0194) | 0.0351 (3, Std:0.0161) |
| WHMMDA Su et al. (2018; 2015) | 0.1387 (3, Std:0.0463) | 0.1654 (3, Std:0.0469) | 0.1236 (3, Std:0.0509) |
| $\ell_{1,2}$ LDA Nie et al. (2021b) | 0.0416 (2, Std = 0.0231) | 0.0916 (2, Std = 0.0174) | 0.0516 (1, Std = 0.0130) |
| MM4MM (QP-MMDA) Omati et al. (2025) | 0.1220 (3, Std:0.0607) | 0.1403 (3, Std:0.0611) | 0.0919 (3, Std:0.0562) |
| MMRA Wang et al. (2024) | 0.0485 (3, Std:0.0293) | 0.0769 (3, Std:0.0453) | 0.0300 (3, Std:0.0173) |
| GDMM-QF (PWCRA) | **0.0257** (3, Std:0.0099) | **0.0740** (3, Std:0.0208) | **0.0262** (3, Std:0.0108) |
| Yale Dataset | | | |
| Method | Classifier | | |
| | 1-NN | NM | QDA |
| LDA Fisher (1936); Rao (1948) | 0.0680 (11, Std:0.0329) | 0.0773 (11, Std:0.0295) | 0.3973 (11, Std:0.0910) |
| MMDA Bian & Tao (2011b) | 0.0747 (11, Std:0.0355) | 0.0693 (16, Std:0.0262) | 0.3673 (11, Std:0.0664) |
| WHMMDA Su et al. (2018; 2015) | 0.1987 (46, Std:0.0528) | 0.0627 (46, Std:0.0203) | 0.7020 (6, Std:0.0675) |
| $\ell_{1,2}$ LDA Nie et al. (2021b) | 0.0987 (41, Std = 0.0170) | **0.0167** (41, Std = 0.0105) | 0.3387 (6, Std = 0.0164) |
| MM4MM (QP-MMDA) Omati et al. (2025) | 0.2293 (46, Std:0.0513) | 0.0920 (46, Std:0.0387) | 0.7553 (6, Std:0.0592) |
| MMRA Wang et al. (2024) | 0.9011 (46, Std:0.06) | 0.8452 (46, Std:0.0301) | 0.7839 (46, Std:0.101) |
| GDMM-QF (PWCRA) | **0.0287** (11, Std:0.0117) | 0.0240 (11, Std:0.0165) | **0.2573** (11, Std:0.0424) |

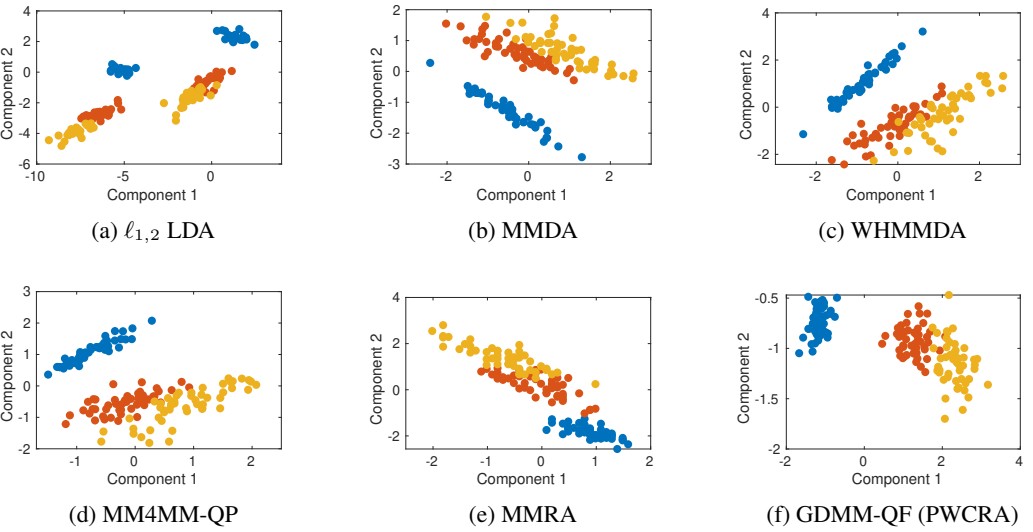

(a) $\ell_{1,2}$ LDA      (b) MMDA      (c) WHMMDA

(d) MM4MM-QP      (e) MMRA      (f) GDMM-QF (PWCRA)

Figure 1: A comparison of results from six different methods.

## 4.3 RESULTS AND ANALYSIS

In this section, we analyze the performance of our proposed method, GDMM-QF (PWCRA), against the baseline algorithms. The evaluation is based on classification error rates from Table 1 and a visual analysis of class separability from Figure 1. Additionally, we have provided a computational analysis; the detailed results are accessible in Appendix H.

We begin by examining the classification performance on the **COIL-20 dataset**. As shown in Table 1, our GDMM-QF (PWCRA) method achieves the lowest mean error rates across all three classifiers. With the 1-NN classifier, it obtains an error of **0.0026**, which is substantially better than the next best methods, LDA (0.0046) and MMRA (0.0051). This trend continues for the NM and QDA classifiers, where our approach also secures the top position. Furthermore, the standard deviation for our method is consistently among the lowest, indicating more stable and reliable performance over the 20 independent trials.

Next, we turn to the **Diamond dataset**. Here again, GDMM-QF (PWCRA) demonstrates its superiority by a significant margin. It achieves the lowest error rates for 1-NN (0.0257), NM (0.0740), and QDA (0.0262). Its performance is particularly noteworthy when compared to methods like WH-MMDA and MM4MM (QP-MMDA), which appears to fail entirely. The error rate of our method with the 1-NN classifier is nearly half that of its closest competitors, LDA, $\ell_{1,2}$ LDA, and MMDA, underscoring its effectiveness on this type of data.

The analysis continues with the **Yale Face Dataset**, a highly challenging task due to variations in lighting and facial expression. GDMM-QF (PWCRA) once again delivers a standout performance. It achieves the lowest error rate with the 1-NN classifier at **0.0287**, more than halving the error of the next best method, LDA (0.0680). It also secures the best result with the QDA classifier. While $\ell_{1,2}$ LDA obtains the top score for the NM classifier, our method's result is highly competitive and a close second. In contrast, several competing methods, including MMRA, WHMMDA, and MM4MM (QP-MMDA), perform very poorly, with error rates often exceeding 70%. This highlights the robustness of our algorithm in handling complex, real-world variations where other methods fail.

Overall, across the nine experimental settings (three datasets and three classifiers), the proposed GDMM-QF (PWCRA) method ranks first in eight of them. This consistent, top-tier performance provides strong quantitative evidence of its superior ability to find highly discriminative low-dimensional subspaces.

To provide a qualitative perspective, we now analyze the 2D projections of the Iris dataset shown in Figure 1. These plots visualize how well each method separates the three classes. The projections generated by the competitors—LDA (Fig. 1a), MMDA (Fig. 1b), WHMMDA (Fig. 1c), MM4MM-QP (Fig. 1d), and MMRA (Fig. 1e)—show limited success. While they separate one class (blue points), the other two classes (green and red points) remain significantly **overlapped**. In several cases, such as with MMDA and MMRA, the projected points for these two classes also exhibit **high internal variance**, meaning the points of the same class are widely scattered. This high intra-class scatter and inter-class overlap create a decision boundary that is ambiguous and complex, which is a major disadvantage as it directly leads to higher misclassification rates. In stark contrast, the projection from our **GDMM-QF (PWCRA)** method (Fig. 1f) demonstrates a markedly superior outcome. It produces three well-separated and compact clusters with clear margins between them. Our method not only pushes the class clusters apart but also minimizes the internal variance within each class. This leads to a low-dimensional space where classes are linearly separable with high confidence, explaining the superior quantitative results observed in our experiments.

In summary, the step-by-step analysis of both the quantitative error rates and the qualitative visualizations confirms the exceptional performance of the proposed GDMM-QF (PWCRA) algorithm. It consistently outperforms established methods in finding subspaces that yield better class separability and lower classification error.

## 5 CONCLUSION

In this work, we presented a solution to the problem of worst-case class separation in discriminative dimensionality reduction. We investigated an objective based on maximizing the minimum pairwise ratio of between-class to within-class scatter, leading to the development of GDMM-QF, a robust two-level optimization algorithm. By combining a generalized Dinkelbach procedure with a custom minorization-maximization (MM) solver, GDMM-QF efficiently solves the underlying non-convex problem without requiring hyperparameter tuning. Our investigation established that the algorithm is provably convergent and computationally efficient. Finally, our experimental validation on several benchmark datasets substantiated the effectiveness of this approach, demonstrating its consistent ability to outperform leading state-of-the-art methods in classification accuracy.

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

APPENDICES

## A  THE MINORIZATION-MAXIMIZATION (MM) PRINCIPLE

### A.1  THE GENERAL MM FRAMEWORK

The minorization-maximization (MM) algorithm is an iterative optimization technique for solving constrained maximization problems of the form:

$$\max_{\mathbf{T} \in \chi} f(\mathbf{T}), \tag{20}$$

where $f(\mathbf{T})$ is the objective function, $\mathbf{T}$ is the optimization variable, and $\chi$ represents the feasible set. The core principle of the MM algorithm involves iteratively solving a sequence of simpler optimization problems. Specifically, at each iteration $t$, a surrogate function $g(\mathbf{T} \mid \mathbf{T}^t)$, termed a minorizer of $f(\mathbf{T})$, is constructed. This surrogate must satisfy two fundamental conditions:

$$g(\mathbf{T} \mid \mathbf{T}^t) \leq f(\mathbf{T}), \quad \forall \mathbf{T} \in \chi, \tag{21}$$
$$g(\mathbf{T}^t \mid \mathbf{T}^t) = f(\mathbf{T}^t). \tag{22}$$

The first condition (21) ensures that the surrogate function provides a global **lower bound** for the original objective function. The second condition (22) guarantees that the surrogate is tangent to (or "touches") the objective function at the current iterate $\mathbf{T}^t$.

The subsequent iterate, $\mathbf{T}^{t+1}$, is then obtained by **maximizing** this surrogate function over the feasible set:

$$\mathbf{T}^{t+1} \in \arg\max_{\mathbf{T} \in \chi} g\left(\mathbf{T} \mid \mathbf{T}^t\right). \tag{23}$$

This iterative process, encompassing the construction and maximization of the surrogate, is repeated until a convergence criterion is met, typically when the relative change in the objective function value falls below a predefined tolerance $\epsilon$.

A key property of the MM algorithm is the guaranteed monotonic improvement of the objective function value at each step. This **ascent property** is readily established through the following sequence of inequalities:

$$f(\mathbf{T}^{t+1}) \geq g(\mathbf{T}^{t+1} \mid \mathbf{T}^t) \geq g(\mathbf{T}^t \mid \mathbf{T}^t) = f(\mathbf{T}^t). \tag{24}$$

The first inequality holds due to the surrogate condition in (21), the second follows from the maximization step in (23), and the final equality is a direct consequence of the tangency condition in (22). This guarantees that the sequence of objective values $\{f(\mathbf{T}^t)\}$ is non-decreasing.

### A.2  AN MM FRAMEWORK FOR MAX-MIN PROBLEMS (MM4MM)

The MM principle can be effectively extended to address max-min optimization problems, which are structured as:

$$\max_{\mathbf{T} \in \mathcal{X}} \left\{ f(\mathbf{T}) \triangleq \min_{i=1,\ldots,K} f_i(\mathbf{T}) \right\}, \tag{25}$$

where the overall objective $f(\mathbf{T})$ is defined by the pointwise minimum of a set of functions $\{f_i(\mathbf{T})\}_{i=1}^{K}$. To solve this problem using an MM approach, we construct a composite surrogate function for $f(\mathbf{T})$.

Let $g_i(\mathbf{T} \mid \mathbf{T}^t)$ be a valid minorizer for each individual function $f_i(\mathbf{T})$, satisfying the standard MM conditions:

$$g_i(\mathbf{T} \mid \mathbf{T}^t) \leq f_i(\mathbf{T}), \tag{26}$$
$$g_i(\mathbf{T}^t \mid \mathbf{T}^t) = f_i(\mathbf{T}^t). \tag{27}$$

A natural choice for the overall surrogate function $g(\mathbf{T} \mid \mathbf{T}^t)$ is the pointwise minimum of the individual surrogates:

$$g\left(\mathbf{T} \mid \mathbf{T}^t\right) \triangleq \min_{i=1,\ldots,K} g_i\left(\mathbf{T} \mid \mathbf{T}^t\right). \tag{28}$$

It can be verified that this construction yields a valid minorizer for the max-min objective $f(\mathbf{T})$. The lower-bound property is established as follows:

$$g(\mathbf{T} \mid \mathbf{T}^t) = \min_i g_i(\mathbf{T} \mid \mathbf{T}^t) \le \min_i f_i(\mathbf{T}) = f(\mathbf{T}), \tag{29}$$

and the tangency condition is similarly met:

$$g(\mathbf{T}^t \mid \mathbf{T}^t) = \min_i g_i(\mathbf{T}^t \mid \mathbf{T}^t) = \min_i f_i(\mathbf{T}^t) = f(\mathbf{T}^t). \tag{30}$$

By applying the standard MM update rule with the surrogate defined in (28), the resulting sequence of iterates $\{\mathbf{X}^t\}$ is guaranteed to monotonically increase the max-min objective function and converge to a stationary point. For a more detailed exposition of the MM approach and its applications, including techniques for constructing surrogate functions, we refer the reader to Sun et al. (2017); Saini et al. (2024).

## B  PROOF OF THEOREM 1

*Proof.* Supposing $\mathbf{T}^*$ and $\lambda^*$ are the optimal solution and corresponding objective function value of problem (2), then the following holds:

$$\min_{1 \le i < j \le C} \frac{f_{ij}(\mathbf{T}^*)}{g_{ij}(\mathbf{T}^*)} = \lambda^*.$$

Moreover, for any feasible solution $\mathbf{T} \in \mathcal{X}$, since $g_{ij}(\mathbf{T}) > 0$, we have:

$$\min_{1 \le i < j \le C} \frac{f_{ij}(\mathbf{T})}{g_{ij}(\mathbf{T})} \le \lambda^* \implies \min_{1 \le i < j \le C} (f_{ij}(\mathbf{T}) - \lambda^* g_{ij}(\mathbf{T})) \le 0.$$

So we can determine that:

$$h(\lambda^*) = \max_{\mathbf{T} \in \mathcal{X}} \min_{1 \le i < j \le C} (f_{ij}(\mathbf{T}) - \lambda^* g_{ij}(\mathbf{T})) \le 0.$$

On the other hand, for the optimal solution $\mathbf{T}^*$:

$$\min_{1 \le i < j \le C} \frac{f_{ij}(\mathbf{T}^*)}{g_{ij}(\mathbf{T}^*)} = \lambda^* \implies \min_{1 \le i < j \le C} (f_{ij}(\mathbf{T}^*) - \lambda^* g_{ij}(\mathbf{T}^*)) = 0.$$

Thus, we can obtain $h(\lambda^*) = 0$. That is, the optimal function value $\lambda^*$ of the problem in (2) is the root of the function $h(\lambda)$ defined in Theorem 1. This completes the proof of Theorem 1. □

## C  PROOF OF THEOREM 2

*Proof.* Algorithm 1 can be interpreted as an application of Newton's method to find the root of the function $h(\lambda) = \max_{\mathbf{T} \in \mathcal{X}} \min_{ij} (f_{ij}(\mathbf{T}) - \lambda g_{ij}(\mathbf{T}))$. The first-order Taylor expansion of $h(\lambda)$ around the current estimate $\lambda_k$ is given by:

$$h(\lambda) \approx h(\lambda_k) + h'(\lambda_k)(\lambda - \lambda_k).$$

The derivative of $h(\lambda)$ with respect to $\lambda$ is $h'(\lambda) = -g_{ab}(\mathbf{T}^{k+1})$, where $\mathbf{T}^{k+1}$ is the argument that maximizes the inner expression for a given $\lambda$, and $(a, b)$ is the index pair corresponding to the minimum value for that $\mathbf{T}^{k+1}$. Newton's method finds the root by setting this linear approximation to zero:

$$0 = h(\lambda_k) - g_{ab}(\mathbf{T}^{k+1})(\lambda - \lambda_k).$$

Solving for $\lambda$ yields the update rule for the next iterate, which we denote $\lambda_{k+1}$:

$$\lambda_{k+1} = \lambda_k + \frac{h(\lambda_k)}{g_{ab}(\mathbf{T}^{k+1})} = \lambda_k + \frac{f_{ab}(\mathbf{T}^{k+1}) - \lambda_k g_{ab}(\mathbf{T}^{k+1})}{g_{ab}(\mathbf{T}^{k+1})} = \frac{f_{ab}(\mathbf{T}^{k+1})}{g_{ab}(\mathbf{T}^{k+1})}.$$

This formulation is precisely the update rule presented in Step 2 of Algorithm 1. Thus, the algorithm implements Newton's method to solve $h(\lambda) = 0$. □

# D    PROOF OF THEOREM 3

*Proof.* **Monotonic Convergence:** In the $k$-th iteration of Algorithm 1, let $\lambda_k = \frac{f_{ab}(\mathbf{T}^k)}{g_{ab}(\mathbf{T}^k)} = \min_{ij} \frac{f_{ij}(\mathbf{T}^k)}{g_{ij}(\mathbf{T}^k)}$. This implies that $f_{ij}(\mathbf{T}^k) - \lambda_k g_{ij}(\mathbf{T}^k) \geq 0$ for all pairs $(i, j)$, and consequently, $\min_{ij}(f_{ij}(\mathbf{T}^k) - \lambda_k g_{ij}(\mathbf{T}^k)) = 0$. The subproblem solved in Step 3 yields $\mathbf{T}^{k+1}$, which defines the value of $h(\lambda_k)$:

$$h(\lambda_k) = f_{cd}(\mathbf{T}^{k+1}) - \lambda_k g_{cd}(\mathbf{T}^{k+1}) = \max_{\mathbf{T} \in \mathcal{X}} \min_{ij}(f_{ij}(\mathbf{T}) - \lambda_k g_{ij}(\mathbf{T})),$$

where $(c, d)$ is the index pair corresponding to the minimum value for the solution $\mathbf{T}^{k+1}$. Since $\mathbf{T}^k$ is a feasible candidate for this maximization, we must have:

$$h(\lambda_k) \geq \min_{ij}(f_{ij}(\mathbf{T}^k) - \lambda_k g_{ij}(\mathbf{T}^k)) = 0.$$

The inequality $f_{cd}(\mathbf{T}^{k+1}) - \lambda_k g_{cd}(\mathbf{T}^{k+1}) \geq 0$ directly leads to $\frac{f_{cd}(\mathbf{T}^{k+1})}{g_{cd}(\mathbf{T}^{k+1})} \geq \lambda_k$. As the next iterate is defined as $\lambda_{k+1} = \min_{ij} \frac{f_{ij}(\mathbf{T}^{k+1})}{g_{ij}(\mathbf{T}^{k+1})}$, and since $\lambda_{k+1}$ is the minimum of all such ratios, we know that $\frac{f_{cd}(\mathbf{T}^{k+1})}{g_{cd}(\mathbf{T}^{k+1})} \geq \lambda_{k+1}$. Combining these, we have established that $\lambda_{k+1} \geq \lambda_k$. This proves that the objective value is non-decreasing in each iteration of Algorithm 1.

**Global Optimality:** Suppose the algorithm converges at iteration $k$, which means $\lambda_k = \lambda_{k+1}$. Such convergence is guaranteed since the sequence $\{\lambda_k\}$ is monotonically non-decreasing (as shown above) and bounded above: for any $\mathbf{T} \in \mathcal{S}$ and pair $(i, j)$, the ratio $\frac{f_{ij}(\mathbf{T})}{g_{ij}(\mathbf{T})} \leq \frac{\lambda_{\max}(\tilde{\mathbf{S}}_C^{ij})}{\lambda_{\min}(\tilde{\mathbf{S}}_W^{ij})}$ by the Rayleigh-Ritz theorem and the constraint $\mathbf{T}^T \mathbf{T} = \mathbf{I}_m$, thus $f(\mathbf{T})$ is bounded by $M := \max_{ij} \frac{\lambda_{\max}(\tilde{\mathbf{S}}_C^{ij})}{\lambda_{\min}(\tilde{\mathbf{S}}_W^{ij})}$; therefore, by the monotone convergence theorem, the sequence converges to some limit $\lambda^*$. From the update rule, this implies $\lambda_k = \min_{ij} \frac{f_{ij}(\mathbf{T}^{k+1})}{g_{ij}(\mathbf{T}^{k+1})}$. Let $(c, d)$ be the index pair for which this minimum is achieved for $\mathbf{T}^{k+1}$. Then $\lambda_k = \frac{f_{cd}(\mathbf{T}^{k+1})}{g_{cd}(\mathbf{T}^{k+1})}$, which can be rearranged to $f_{cd}(\mathbf{T}^{k+1}) - \lambda_k g_{cd}(\mathbf{T}^{k+1}) = 0$. This is equivalent to stating that $h(\lambda_k) = 0$. By Theorem 1, a solution $\lambda^*$ is optimal if and only if $h(\lambda^*) = 0$. Since the converged solution $\lambda_k$ satisfies this condition, it is the globally optimal solution. We can formalize this by contradiction: assume convergence occurs but $h(\lambda_k) > 0$. This would imply $f_{cd}(\mathbf{T}^{k+1}) - \lambda_k g_{cd}(\mathbf{T}^{k+1}) > 0$, leading to $\lambda_{k+1} = \min_{ij} \frac{f_{ij}(\mathbf{T}^{k+1})}{g_{ij}(\mathbf{T}^{k+1})} > \lambda_k$, which contradicts the convergence assumption $\lambda_k = \lambda_{k+1}$. Thus, the algorithm must converge to the global optimum. $\square$

# E    THE PROOF OF LEMMA 1

*Proof.* Consider the trace term $\mathrm{tr}\left(\mathbf{T}^T \tilde{\mathbf{S}}_{Cij} \mathbf{T}\right)$ for any pair $(i, j)$. Using the cyclic property of the trace, we can write:

$$\mathrm{tr}\left(\mathbf{T}^T \tilde{\mathbf{S}}_{Cij} \mathbf{T}\right) = \mathrm{tr}\left(\tilde{\mathbf{S}}_{Cij} \mathbf{T}\mathbf{T}^T\right). \tag{31}$$

Furthermore, for any orthogonal matrix $\mathbf{Q}$ (where $\mathbf{Q}\mathbf{Q}^T = \mathbf{I}$), we can insert it into the expression without changing its value:

$$\mathrm{tr}\left(\tilde{\mathbf{S}}_{Cij} \mathbf{T}\mathbf{T}^T\right) = \mathrm{tr}\left(\tilde{\mathbf{S}}_{Cij} \mathbf{T}\mathbf{Q}\mathbf{Q}^T\mathbf{T}^T\right). \tag{32}$$

Let $\mathbf{T}$ be any matrix that satisfies the relaxed constraint $\mathbf{T}^T \mathbf{T} \preccurlyeq \mathbf{I}$. We can always choose a specific orthogonal matrix $\mathbf{Q}$ such that it diagonalizes $\mathbf{T}^T \mathbf{T}$, a result from the singular value decomposition. This gives $\mathbf{Q}^T \mathbf{T}^T \mathbf{T} \mathbf{Q} = \mathbf{\Lambda}$, where $\mathbf{\Lambda}$ is a diagonal matrix of the eigenvalues of $\mathbf{T}^T \mathbf{T}$. The constraint $\mathbf{T}^T \mathbf{T} \preccurlyeq \mathbf{I}$ ensures that these eigenvalues satisfy $\mathbf{\Lambda}_{kk} \leq 1$ for all $k$.

Since the matrices $\mathbf{T}^T \mathbf{T}$ and $\mathbf{T}\mathbf{T}^T$ share the same non-zero eigenvalues, it follows that the eigendecomposition of $\mathbf{T}\mathbf{T}^T$ is given by:

$$\mathbf{T}\mathbf{T}^T = \mathbf{V}\mathbf{\Lambda}\mathbf{V}^T, \tag{33}$$

where $\mathbf{V}$ contains the principal eigenvectors of $\mathbf{T}\mathbf{T}^T$ and satisfies $\mathbf{V}^T\mathbf{V} = \mathbf{I}$. Using (33), we can expand the trace term:

$$\text{tr}\left(\mathbf{T}^T\tilde{\mathbf{S}}_{Cij}\mathbf{T}\right) = \text{tr}\left(\tilde{\mathbf{S}}_{Cij}\mathbf{V}\mathbf{\Lambda}\mathbf{V}^T\right) = \text{tr}\left((\mathbf{V}^T\tilde{\mathbf{S}}_{Cij}\mathbf{V})\mathbf{\Lambda}\right) \tag{34}$$

$$= \sum_{k=1}^{m}\left(\mathbf{V}^T\tilde{\mathbf{S}}_{Cij}\mathbf{V}\right)_{kk}\mathbf{\Lambda}_{kk} \tag{35}$$

$$\leq \sum_{k=1}^{m}\left(\mathbf{V}^T\tilde{\mathbf{S}}_{Cij}\mathbf{V}\right)_{kk}, \tag{36}$$

where the inequality in (36) holds because $(\mathbf{V}^T\tilde{\mathbf{S}}_{Cij}\mathbf{V})_{kk} \geq 0$ and $\mathbf{\Lambda}_{kk} \leq 1$.

The inequality shows that the objective function takes its maximum value when $\mathbf{\Lambda}_{kk} = 1$ for all $k$, which corresponds to $\mathbf{\Lambda} = \mathbf{I}$. This indicates that the global maximizer of the problem under the relaxed constraint $\mathbf{T}^T\mathbf{T} \preccurlyeq \mathbf{I}$ must inherently satisfy the original, stricter constraint $\mathbf{T}^T\mathbf{T} = \mathbf{I}$. Therefore, the relaxation does not alter the solution, completing the proof of Lemma 1. $\qquad\square$

## F PROOF OF (19)

*Proof.* Our objective is to find an explicit expression for the matrix $\mathbf{Q}$ in the equality $\text{tr}\left(\mathbf{A}^T\mathbf{A}\mathbf{\Phi}^t\right) = \mathbf{z}^T\mathbf{Q}\mathbf{z}$. We begin by re-indexing the terms. Let $\tilde{\mathbf{A}}_k = \mathbf{A}_{ij}$ and $\tilde{z}_k = z_{ij}$, where the index $k$ corresponds to the pair $(i,j)$ for $1 \leq i < j \leq C$ via the mapping $k = \frac{(i-1)(2C-i)}{2} + j - i$. The total number of such indices is $K = \frac{C(C-1)}{2}$. The matrix $\mathbf{A}$ can then be written as $\mathbf{A} = \sum_{k=1}^{K}\tilde{z}_k\tilde{\mathbf{A}}_k$.

First, we expand the term $\mathbf{A}^T\mathbf{A}$:

$$\mathbf{A}^T\mathbf{A} = \left(\sum_{k=1}^{K}\tilde{z}_k\tilde{\mathbf{A}}_k\right)^T\left(\sum_{l=1}^{K}\tilde{z}_l\tilde{\mathbf{A}}_l\right) = \sum_{k=1}^{K}\sum_{l=1}^{K}\tilde{z}_k\tilde{z}_l\tilde{\mathbf{A}}_k^T\tilde{\mathbf{A}}_l.$$

Taking the trace after right-multiplying by $\mathbf{\Phi}^t$ yields:

$$\text{tr}\left(\mathbf{A}^T\mathbf{A}\mathbf{\Phi}^t\right) = \sum_{k=1}^{K}\sum_{l=1}^{K}\tilde{z}_k\tilde{z}_l\,\text{tr}\left(\tilde{\mathbf{A}}_k^T\tilde{\mathbf{A}}_l\mathbf{\Phi}^t\right). \tag{37}$$

The quadratic form $\mathbf{z}^T\mathbf{Q}\mathbf{z}$ can be expanded as:

$$\mathbf{z}^T\mathbf{Q}\mathbf{z} = \sum_{k=1}^{K}\sum_{l=1}^{K}\tilde{z}_k Q_{k,l}\tilde{z}_l. \tag{38}$$

By equating the coefficients of $\tilde{z}_k\tilde{z}_l$ in (37) and (38), we identify the entries of $\mathbf{Q}$ as:

$$Q_{k,l} = \text{tr}\left(\tilde{\mathbf{A}}_k^T\tilde{\mathbf{A}}_l\mathbf{\Phi}^t\right).$$

To express $\mathbf{Q}$ in a compact matrix form, we use a property of the Kronecker product $\otimes$. Let $\boldsymbol{v}_k = \text{vec}(\tilde{\mathbf{A}}_k)$. The trace term can be written as:

$$\text{tr}\left(\tilde{\mathbf{A}}_k^T\tilde{\mathbf{A}}_l\mathbf{\Phi}^t\right) = \boldsymbol{v}_k^T\left(\mathbf{I}\otimes\mathbf{\Phi}^t\right)\boldsymbol{v}_l,$$

Substituting this into the sum gives:

$$\text{tr}\left(\mathbf{A}^T\mathbf{A}\mathbf{\Phi}^t\right) = \sum_{k=1}^{K}\sum_{l=1}^{K}\tilde{z}_k\left(\boldsymbol{v}_k^T\left(\mathbf{I}\otimes\mathbf{\Phi}^t\right)\boldsymbol{v}_l\right)\tilde{z}_l.$$

Let us construct a matrix $\mathbf{S}$ by stacking the vectors $\boldsymbol{v}_k$ as its columns: $\mathbf{S} = [\boldsymbol{v}_1, \boldsymbol{v}_2, \ldots, \boldsymbol{v}_K]$. The expression above can then be rewritten as a matrix-vector product:

$$\text{tr}\left(\mathbf{A}^T\mathbf{A}\mathbf{\Phi}^t\right) = \mathbf{z}^T\mathbf{S}^T\left(\mathbf{I}\otimes\mathbf{\Phi}^t\right)\mathbf{S}\mathbf{z}. \tag{39}$$

Comparing (39) with $\mathbf{z}^T\mathbf{Q}\mathbf{z}$, we deduce the final form of $\mathbf{Q}$:

$$\mathbf{Q} = \mathbf{S}^T\left(\mathbf{I}\otimes\mathbf{\Phi}^t\right)\mathbf{S},$$

where $\mathbf{S} = [\boldsymbol{v}_1, \boldsymbol{v}_2, \ldots, \boldsymbol{v}_K]$ and $\boldsymbol{v}_k = \text{vec}(\mathbf{A}_{ij})$ with $k = \frac{(i-1)(2C-i)}{2} + j - i$. The proof is complete. $\qquad\square$

---

**Algorithm 1** Outer Loop: Generalized Dinkelbach Algorithm for PWCRA

---

1: **Initialize:** Feasible $\mathbf{T}^0$, set $k = 0$, tolerance $\epsilon$.
2: **while** not converged **do**
3:     Compute worst-case ratio: $\lambda_k = \min_{1 \leq i < j \leq C} \frac{\text{tr}\left((\mathbf{T}^k)^T \mathbf{S}_b^{ij} \mathbf{T}^k\right)}{\text{tr}\left((\mathbf{T}^k)^T \mathbf{S}_w^{ij} \mathbf{T}^k\right)}$.
4:     Solve the max-min subproblem for the next iterate:

$$\mathbf{T}^{k+1} = \arg \max_{\mathbf{T}^T \mathbf{T} = \mathbf{I}_m} \min_{1 \leq i < j \leq C} \left\{ \text{tr}(\mathbf{T}^T (\mathbf{S}_b^{ij} - \lambda_k \mathbf{S}_w^{ij}) \mathbf{T}) \right\}.$$

5:     Check for convergence (e.g., if $|\lambda_k - \lambda_{k-1}| < \epsilon$).
6:     Increment $k \leftarrow k + 1$.
7: **end while**
8: **Output:** Optimal transformation $\mathbf{T}^* = \mathbf{T}^k$.

---

**Algorithm 2** Inner Loop: Solving the Max-Min Subproblem via MM (SDP Approach)

---

1: **Input:** Initial estimate $\mathbf{T}^0$, matrices $\{\tilde{\mathbf{S}}_{Cij}\}$, and convergence threshold $\epsilon$.
2: **Initialize:** Set $t = 0$.
3: **repeat**
4:     Compute coefficients $\{\mathbf{A}_{ij}, c_{ij}\}$ using $\mathbf{T}^t$ via (8) and (9).
5:     Solve for the optimal weights $\mathbf{z}^*$ by solving the convex problem (12).
6:     Compute the updated projection matrix $\mathbf{T}^{t+1}$ using $\mathbf{z}^*$ via (13).
7:     Increment $t \leftarrow t + 1$.
8: **until** a stopping criterion is met (e.g., $\frac{\|\mathbf{T}^t - \mathbf{T}^{t-1}\|_F}{\|\mathbf{T}^{t-1}\|_F} \leq \epsilon$)
9: **Output:** Optimal projection matrix $\mathbf{T}^* = \mathbf{T}^t$.

---

# G ALGORITHMS

The algorithms below detail our nested optimization strategy. **Algorithm 1** describes the outer loop, which applies a generalized Dinkelbach method Dinkelbach (1967) to transform the original fractional problem into a sequence of simpler subproblems. **Algorithm 2** shows how to solve this subproblem using a minorization-maximization (MM) approach that results in a semidefinite program. Finally, **Algorithm 3** presents our proposed and much faster inner-loop solver, which solves the dual of the SDP via an efficient Quadratic Program.

**Algorithm 3** Inner Loop: Solving the Max-Min Subproblem via MM (QP Approach)

---

1: **Input:** Initial weights $\mathbf{z}^0$, problem data $\{c_{ij}, \mathbf{A}_{ij}\}$, convergence tolerance $\epsilon > 0$.
2: **Initialize:** Set iteration counter $t = 0$.
3: **repeat**
4:  **Update Auxiliary Matrix:** Compute $\mathbf{\Phi}^t$ based on the current $\mathbf{z}^t$:

$$\mathbf{\Phi}^t = \left( \mathbf{A}(\mathbf{z}^t)^T \mathbf{A}(\mathbf{z}^t) \right)^{-\frac{1}{2}}$$

5:  **Solve QP Subproblem:** Update the weights by solving the quadratic program from (18) to find $\mathbf{z}^{t+1}$.
6:  Increment iteration counter: $t \leftarrow t + 1$.
7: **until** the relative change in $\mathbf{z}$ is below the tolerance: $\frac{\|\mathbf{z}^t - \mathbf{z}^{t-1}\|}{\|\mathbf{z}^{t-1}\|} \leq \epsilon$
8: **Output:** The converged weight vector $\mathbf{z}^* = \mathbf{z}^t$.

---

# H    ADDITIONAL RESULTS

Table 2: Mean Runtime per Iteration (in seconds) on COIL-20, Diamond, and Yale Datasets.

| **COIL-20 Dataset** | | | | |
|---|---|---|---|---|
| Method | Dimension | | | |
| | 1 | 12 | 25 | 49 |
| LDA Fisher (1936); Rao (1948) | $1.4123 \pm 0.1141$ | $0.4725 \pm 0.0684$ | N/A | N/A |
| MMDA Bian & Tao (2011b) | $53.1535 \pm 2.7430$ | $52.9179 \pm 6.9516$ | $47.1447 \pm 5.4055$ | $40.5701 \pm 5.1543$ |
| WHMMDA Su et al. (2018; 2015) | $27.5446 \pm 1.0775$ | $30.3072 \pm 1.3726$ | $26.6994 \pm 1.3850$ | $26.4311 \pm 2.4698$ |
| $\ell_{1,2}$ LDA Nie et al. (2021b) | $0.0513 \pm 0.0098$ | $0.0220 \pm 0.0053$ | $0.0192 \pm 0.0085$ | $0.0240 \pm 0.0051$ |
| MMRA Wang et al. (2024) | $2.5164 \pm 0.1602$ | $2.3128 \pm 0.1279$ | $1.9990 \pm 0.1019$ | $2.1361 \pm 0.0860$ |
| GDMM-QF (PWCRA) | $1.6976 \pm 0.1274$ | $1.5403 \pm 0.1116$ | $1.4302 \pm 0.0693$ | $1.4247 \pm 0.0950$ |
| **Diamond Dataset** | | | | |
| Method | Dimension | | | |
| | 1 | 2 | 3 | 4 |
| LDA Fisher (1936); Rao (1948) | $1.2912 \pm 0.0266$ | $0.1276 \pm 0.0332$ | $0.1214 \pm 0.0324$ | N/A |
| MMDA Bian & Tao (2011b) | $5.0111 \pm 0.1612$ | $2.1866 \pm 0.1501$ | $2.2577 \pm 0.1497$ | $1.8733 \pm 0.0876$ |
| WHMMDA Su et al. (2018; 2015) | $2.0015 \pm 0.1024$ | $2.0556 \pm 0.1505$ | $1.9926 \pm 0.1488$ | $1.9105 \pm 0.1284$ |
| $\ell_{1,2}$ LDA Nie et al. (2021b) | $0.0476 \pm 0.0056$ | $0.0543 \pm 0.0024$ | $0.0080 \pm 0.0020$ | $0.0046 \pm 0.0012$ |
| MMRA Wang et al. (2024) | $1.0732 \pm 0.0173$ | $0.8298 \pm 0.0548$ | $0.8398 \pm 0.0494$ | $0.8560 \pm 0.0779$ |
| GDMM-QF (PWCRA) | $1.5270 \pm 0.1014$ | $1.1372 \pm 0.0871$ | $0.8989 \pm 0.0660$ | $0.9744 \pm 0.0626$ |
| **Yale Dataset** | | | | |
| Method | Dimension | | | |
| | 1 | 12 | 25 | 49 |
| LDA Fisher (1936); Rao (1948) | $1.3692 \pm 0.0867$ | $0.4698 \pm 0.1052$ | N/A | N/A |
| MMDA Bian & Tao (2011b) | $33.8620 \pm 0.9662$ | $29.8367 \pm 1.1881$ | $32.7184 \pm 1.8557$ | $31.3374 \pm 2.5932$ |
| WHMMDA Su et al. (2018; 2015) | $22.1406 \pm 0.8274$ | $21.4248 \pm 0.8639$ | $21.6595 \pm 1.5921$ | $22.1489 \pm 2.4855$ |
| $\ell_{1,2}$ LDA Nie et al. (2021b) | $0.0476 \pm 0.0140$ | $0.0190 \pm 0.0043$ | $0.0105 \pm 0.0022$ | $0.0119 \pm 0.0023$ |
| MMRA Wang et al. (2024) | $2.1801 \pm 0.0703$ | $1.9019 \pm 0.1211$ | $1.6293 \pm 0.1157$ | $1.9257 \pm 0.1131$ |
| GDMM-QF (PWCRA) | $1.5407 \pm 0.0692$ | $1.0347 \pm 0.0580$ | $1.1194 \pm 0.0323$ | $1.1408 \pm 0.0577$ |

In this part, we strive to provide additional results by examining the computational efficiency of the competing methods, with runtimes per iteration detailed in Table 2. As observed in the accuracy results (Table 1), each method achieves its optimal performance at a different projection dimension ($d$), and there is no clear pattern linking a specific dimension to peak accuracy across all algorithms. Therefore, comparing runtimes only at each method's individual optimal dimension would not be a fair or direct comparison. To address this, in Table 2, we evaluate the efficiency of all methods across the same set of dimensions to provide a more equitable analysis.

The computational results in Table 2 highlight the efficiency of our proposed GDMM-QF (PWCRA) method across diverse datasets.

On the COIL-20 dataset, our method is exceptionally efficient, with mean runtimes per iteration consistently under 1.70 seconds across all tested dimensions (1.6976s at $d = 1$, 1.5403s at $d = 12$, 1.4302s at $d = 25$, and 1.4247s at $d = 49$). This is notably faster than MMRA, whose runtimes range from 1.9990s to 2.5164s, and it represents a dramatic speed-up compared to the computationally demanding MMDA (ranging from 40.5701s to 53.1535s) and WHMMDA (ranging from 26.4311s to 30.3072s) algorithms.

For the low-dimensional Diamond dataset, our method's efficiency remains highly competitive, with runtimes ranging from 0.8989s to 1.5270s. Here, its speed is comparable to MMRA (ranging from 0.8298s to 1.0732s) and is again significantly faster than the MMDA and WHMMDA algorithms.

This strong performance continues on the Yale dataset. GDMM-QF (PWCRA) exhibits runtimes between 1.0347s and 1.5407s. This is consistently faster than MMRA (ranging from 1.6293s to 2.1801s) and orders of magnitude more efficient than WHMMDA (approx. 21.42s – 22.15s) and MMDA (approx. 29.84s – 33.86s).

Across all three datasets, while a method like $\ell_{1,2}$ LDA is computationally faster due to its formulation, this speed comes at the cost of significantly lower classification accuracy, as seen in Table 1. In

contrast, our GDMM-QF (PWCRA) method strikes an excellent balance, delivering state-of-the-art accuracy with a very reasonable and competitive computational cost, which makes it highly practical for real-world applications.

