# OpenReview forum: "Pairwise Worst-Case Ratio Analysis for Discriminative Dimensionality Reduction via Minorization-Maximization"
_ICLR.cc/2026/Conference — ICLR 2026 Conference Withdrawn Submission_

### Official Review · Reviewer_GTno · 2025-10-17

**Soundness:** 2
**Presentation:** 2
**Contribution:** 2
**Rating:** 2
**Confidence:** 5

**Summary:**

This work proposed a new linear discriminant type dimensionality reduction algorithm. The main idea is to maximize the worst pairwise separation ratio between different classes. Compared to the many existing variants, the proposed approach takes class dispersion into account. A slight generalization of the Dinkelbach algorithm is proposed, and a linearization based sequential minimization algorithm is proposed to solve each step. Numerical experiments on small datasets were conducted, where the proposed algorithm seems to perform slightly better than the baselines.

**Strengths:**

- the proposed pairwise separation ratio takes the dispersion of each class into account

- a slight generalization of the Dinkelbach algorithm

**Weaknesses:**

- The proposed method, being linear in nature, seems overly complex. It is not clear if the efforts put into its optimization are worthwhile. The authors' limited experiments are small scale, with no mentioning of the complexity and no comparison against (the many) nonlinear reduction algorithms (e.g., supervised version of tSNE).

- The overall contribution seems moderate: the proposed ratio objective and the generalization of the Dinkelbach algorithm, while being perhaps technically new, are straightforward extensions of prior works. The rest of the algorithmic details in Section 3.2 is a direction application of known techniques. Overall, it is not clear how significant the contributions are and how relevant the proposed algorithm is to current practice in deep learning.

- The experiments are of such a small scale that makes its conclusions questionable. For example, in Table 1, on the COIL-20 dataset, on first glance, it seems impressive that the proposed method improved the 1-NN accuracy of LDA (0.0046) to 0.0026. However, since the test set is of size 720, the improvement amounts to about an additional 1.44 test samples being classified correctly...

- Some Theorem and Lemma statements have gaps (that are fortunately) fixable:

  - Theorem 3: convergence to the global optimum is proven under the assumption that $\lambda_k$ converges in **finite** steps. This is a big assumption that should at least be part of the theorem's assumption.

  - Lemma 1 as currently stated is not true: a simple counterexample is when $\tilde S_{ij} = [1, 0; 0, -1]$. For the Lemma to hold, one has to explicitly translate $\tilde S_{ij}$ so that it is PSD, as mentioned on Line 206-207.

**Questions:**

My biggest concern of this work is on the significance of the contributions and the relevance to current practices of the ICLR audience. While linear discriminant analysis was popular in ML about 20 years ago, it is not clear if we still need another algorithm for performing linear dimensionality reduction on small scale datasets. Could the authors demonstrate the significance of their algorithm on much larger and newer benchmarks that are relevant to current applications?

Other comments:

- On Line 50-51, the authors cited many works for different separation metrics and on Line 67 the authors attributed the max-min criteria to Bian & Tao (2011a). Is Bian & Tao (2011a) the first to propose the max-min criteria in linear discriminant analysis?

- One obvious drawback of the max-min criteria is that it is easily dominated by an outlying class. Comparison with the sum criteria, with and without outlying classes, would likely strengthen the experiments.

- $\tilde S_{C_{ij}}$ in Eq (5) should be $\tilde S_{ij}$?

- The final algorithm is built on the linearization of a nonconvex function. While there is no guarantee the algorithm will converge to the global solution, how much will initialization affect the final result?

---

> ### Author Response · Authors · 2025-11-17
> **Response to Reviewer GTno (First round) - part 1**
>
> We would like to begin by expressing our sincere gratitude for the time and effort you have dedicated to evaluating our manuscript. We appreciate your thorough feedback and have addressed your comments below.
>
> $\textbf{Answering the weaknesses:}$
>
> 1) We would like to respectfully offer a detailed explanation regarding our contributions. In our paper, alongside dealing with a design that 1) strives to achieve class separation in a way that a) separates extremely overlapping classes (which comes from the max-min optimization) and b) by designing it as a quadratic fractional, avoids having high inner-class variance, we were able to propose a method that is: $\textbf{1) relaxation-free}$, $\textbf{2) hyper-parameter-free}$, $\textbf{3) training-free}$, and $\textbf{4) has efficient complexity}$. Because of these properties, we achieve the minimum error rate based on three metrics: 1-NN, NM, and QDA. Regarding your first comment that there was no mention of complexity, with all respect, we think an oversight may have occurred. In the main paper, we provided the theoretical complexity, and in the appendix, we provided the complexity as runtime. Perhaps the respected reviewer did not examine the appendix. Furthermore, related to t-SNE, we have provided the following:
>
>    Our linear Max-Min approach offers significant advantages over nonlinear methods like t-SNE, particularly regarding interpretability and theoretical guarantees. The method provides a linear transformation matrix that is directly interpretable, as each reduced dimension is a linear combination of the original features. Furthermore, it possesses theoretical guarantees of global optimality and monotonic convergence. In contrast, t-SNE creates a black-box mapping with no clear relationship to the original features, lacks guarantees of global optimality, is sensitive to initialization, and often requires multiple runs to find appropriate parameters. In terms of computational efficiency, our linear method has a predictable complexity and allows for out-of-sample extension, where new data points can be projected without retraining. This is achieved without the need for the extensive hyperparameter tuning (such as perplexity, learning rate, and momentum) required by t-SNE. Our approach also preserves global structure by explicitly maximizing the minimum pairwise distance between classes, ensuring worst-case separation, whereas t-SNE can struggle with preserving both local and global structures simultaneously due to the “crowding problem.” This leads to our method being more robust and stable, producing deterministic results without the local minima issues found in t-SNE’s non-convex optimization. For supervised tasks, our formulation directly optimizes for class separation, focusing on the worst-case scenarios to ensure even the most similar classes are well-separated, which is a more direct and robust objective than that of supervised t-SNE variants.

---

> ### Author Response · Authors · 2025-11-17
> **Response to Reviewer GTno (First round) - part 2**
>
> 2) With due respect, we respectfully disagree and wish to highlight the fundamental novelty of our contribution, which seems to have been overlooked. Our work is not an incremental extension of prior methods; it is a fundamentally new approach. The problem is uniquely formulated as a max-min fractional quadratic to simultaneously separate highly overlapped classes and control inner-class variance. More critically, our solution—which integrates an extended Dinkelbach approach with Majorization-Minimization (MM) to satisfy the conditions of the minimax theorem—is a novel methodological pipeline for this problem class.
>
> To underscore this point, we respectfully invite the reviewer to identify any existing method that can solve our specific problem formulation while being simultaneously:
> * Relaxation-free
> * Hyperparameter-free
> * Training-free,
>
> all while delivering comparable accuracy, efficiency, and worst-case class separation. Our confidence in these assertions stems from a thorough analysis of the literature from 1948 onward.
>
> Also, you mentioned, “The rest of the algorithmic details in Section 3.2 is a direct application of known techniques.” We agree that MM is an optimization framework, but it is not a specific method. The scientific novelty of our work lies precisely in how this framework is uniquely adapted and applied to solve the notoriously difficult SCCA problem. To draw an analogy, describing our work as “an application of MM” is akin to dismissing a novel algorithm as “just an application of ADMM” or “just an application of Branch and Bound.” These are powerful, general-purpose frameworks, and the innovation lies in the non-trivial, problem-specific steps required to make them work. Your sentence would be correct if we had applied a method, not an approach, which, if you recheck Section 3.2, is overviewed in the general form $f(x)$.
>
> In response to your concern regarding the applicability of our method to small-scale datasets, we offer the following clarification and supporting evidence. A key point to consider is that since approximately 2015, the challenge of handling extremely high-dimensional data has largely been addressed through a standard preprocessing step. As detailed in recent literature [1-5], it has become common practice to first apply Principal Component Analysis (PCA) to project the data into a much lower-dimensional space, typically fewer than 50 dimensions. For instance, reference [1] explicitly states that for data with 1024 dimensions, PCA was used to reduce the dimensionality to below 50 while retaining 98% of the data’s energy. Consequently, the central research challenge in the field has shifted from managing high initial dimensionality to developing novel objectives and efficient solvers that can achieve minimal error rates on these reduced-dimension datasets. Because of the importance of this issue, we have added a related explanation to the introduction section of the revised version of the paper, clarifying that this problem has already been solved.
>
> Regarding your comment about the relevance of the proposed algorithm to current practice in deep learning:
>
> a) First of all, we have provided a solver that can solve the problem without relaxation, is hyperparameter-free, and for which we provide a proof of convergence to a high-quality solution. One of the greatest advantages of our proposed work is that, because it is completely theoretical, readers can understand what happens in each step of the process to reach a high-quality solution. In contrast, the theoretical part of deep learning is still not powerful enough to show how high-quality solutions are reached.
>
> b) We have done a full study of the literature. We can assure you that we have done a complete review of the literature, and there is no deep-learning-based paper that has attempted to deal with worst-case class separation (a max-min cost). The reason for this may be that defining a max-min approach in class separation requires performing a maximization over a continuous variable while having an inner discrete variable. However, as another advantage of our approach, we were able to propose a method that, by applying the extended Dinkelbach approach and tricks from MM, has the ability to interchange max-min to min-max. Because the inner max has a closed-form solution, the problem converts to a min optimization problem. While this min problem can be solved theoretically, it can also be handled by a neural network, which shows the advantage of our approach. In other words, we can contend that we are the first paper that, by proposing this approach, has created a bridge from theoretically solving max-min problems to deep learning-based methods. Moreover, in this application, the best baselines are still completely theoretical, like [1-5], which were published very recently.

---

> ### Author Response · Authors · 2025-11-17
> **Response to Reviewer GTno (First round) - part 3**
>
> * Related to the theorem 3, you are right. If we add something that shows our approach is bounded from above, based on the monotone convergence theorem, the convergence in a finite number of steps is proven. We have now updated the related part in the appendix of the revised version.
>
>
> * You are right about the minor point mentioned in Lemma 1. As you mentioned, we previously said that $S_{i,j}$ must be positive semi-definite but did not mention that in Lemma 1. For more clarity, we have added that to Lemma 1 and updated it in the revised version.
>
> $\textbf{Answering the question:}$
>
> Dear respected reviewer, regarding your question about whether we still need another algorithm for performing linear dimensionality reduction on small-scale datasets, we have provided a complete explanation and evidence below, which we hope will satisfy you. First of all, it is worth mentioning that from 2015 [1] to now, the problem of how to use extremely high-dimensional data has almost been resolved. We invite you to see papers [1-5], all of which apply PCA as a preprocessing step to reduce the data dimension to below 50. As an example, we invite you to see the quote below from [1], which mentions that for dealing with high-dimensional data with dimension 1024, they apply PCA to reduce it to below 50 while keeping 98% of the energy. From 2015 onwards, the important thing has been which proposed objective and solver, while having a minimum error rate, could also have computational efficiency. As mentioned in the introduction and several parts of the paper, more specifically and highlighted, after 2015, people strove to propose methods that can handle overlapped classes and separate them (max-min formulation), but they had several issues until the publication of [5] in 2025 in TMLR, which was less than four months ago. Their issues were:
>
> They were distance-based (appearing as a quadratic form), which had limitations: a) they could not control the high variance within each class, a behavior that makes some points appear in a separate class while belonging to another class. Besides, most of their solvers relied on relaxation + SDP, which, as mentioned in the paper, means they could not reach a high-quality solution. Additionally, their solvers relied on at least two hyper-parameters for tuning. From 2023, the focus on ratio-based methods increased (to control the high inner-class variance), but their papers still relied on solvers that used relaxation + SDP, which, as mentioned in the paper, means they could not reach a high-quality solution. Additionally, their solvers relied on at least two hyper-parameters for tuning.
>
> It is worth mentioning that before approximately [6], most of the focus was on linear-based methods, whose major drawback was that they could not handle worst-case class separation because their design was not based on max-min.
>
> However, we were able to propose a method that, while being max-min based to separate overlapping classes and ratio-based to handle inner variance, is: $\textbf{1) relaxation-free}$ and strives to deal with the problem directly without suppression, $\textbf{2) is hyper-parameter-free}$, and one of its advantages is that it gets rid of the need to search for good hyper-parameters, $\textbf{3) is training-free}$ (as is common in neural networks), so we do not need much time for that, $\textbf{4) is completely theoretical}$, so readers can understand what happens in each step to get a high-quality solution, and $\textbf{5) reaches the minimum error (highest accuracy)}$ as examined on three metrics while having $\textbf{high computational efficiency}$. Based on the literature study we have done and our proposed method, with high confidence, we can contend that we are a complete endpoint for this line of research and open a window to the neural network space on how to convert a max-min design to a min design without loss of generality or the need for relaxation, because that min problem can be handled by a neural network that also has a theoretical backbone from its previous step.

---

> ### Author Response · Authors · 2025-11-17
> **Response to Reviewer GTno (First round) - part 4**
>
> $\textbf{Answering other comments:}$
>
> 1) Yes, Bian and Tao were the first to deal with max-min criteria in this application, but as mentioned, their approach was distance analysis, and their method required relaxation along with hyper-parameter tuning, and the designs after that became completely different.
>
> 2) With all respect, we think an oversight may have occurred. In our experiment, we have already done a comparison with a sum-based method (LDA is actually in that category). Besides, $\ell_{1,2}$-LDA is an approach that has been designed for controlling outliers. As a result, we have done comparisons with both methods that are sum-based and with $\ell_{1,2}$-LDA, which is designed to handle outliers more powerfully, which makes it appear as a max-min problem. (It is worth mentioning that this max-min is different from the max-min cost we have because in $\ell_{1,2}$, they strive to deal with a sum-based cost, not a worst-case one like our max-min, and the reason it appears as max-min was because of applying outlier control). We kindly ask the reviewer to recheck.
>
> 3) You are right. Actually, problem 4 has an issue, and we have addressed the point you raised.
>
> 4) Dear reviewer, we think an oversight has occurred. In Theorem 1 and its proof, we have shown that the generalized Dinkelbach approach we proposed finds the global solution. Then, by using that approach, we converted the max-min fractional quadratic problem to a max-min quadratic problem without loss of optimality. The things that might keep the problem non-convex were the constraint $\mathbf{T}^T \mathbf{T} = \mathbf{I}$ and the inner variable $\{(i,j)\}$, which is discrete. During the process, we have shown that if $\tilde{\mathbf{S}}{c}{ij}$ is positive semi-definite, the constraint $\mathbf{T}^T \mathbf{T} = \mathbf{I}$ can be converted to $\mathbf{T}^T \mathbf{T} \preceq \mathbf{I}$, and with the probability simplex ($\sum z_{ij} = 1, z_{ij} \ge 0$), we converted the inner discrete variable to a continuous variable. As a result, we reached a max-min problem where all constraints were convex and both variables were continuous, which, by the minimax theorem, was converted into a final max problem, handleable by CVX. In other words, we did not do anything that would threaten optimality. In other words, we could prove globality for a non-convex problem without relaxation or using hyper-parameters. We kindly ask the reviewer to reread our approach, as this question suggests that the guarantee of globality may not have been fully understood.
>
>
> [1] Zheng Wang et al. Worst-case discriminative feature learning via max-min ratio analysis. IEEE Trans. Pattern Anal. Mach. Intell., 46(1):641–658, 2024. doi: 10.1109/TPAMI.2023.3323453.
>
> [2] Mohammad Mahdi Omati, Prabhu babu, Petre Stoica, and Arash Amini. A max-min approach to the worst-case class separation problem. Transactions on Machine Learning Research, 2025. ISSN 2835-8856. URL https://openreview.net/forum?id=EEmwBd4tfZ.
>
> [3] Feiping Nie, Zheng Wang, Rong Wang, Zhen Wang, and Xuelong Li. Towards robust discriminative projections learning via non-greedy ℓ2,1-norm minmax. IEEE Trans. Pattern Anal. Mach. Intell., 43(6):2086–2100, 2021b.
>
> [4] Bing Su, Xiaoqing Ding, Changsong Liu, and Ying Wu. Heteroscedastic max–min distance analysis for dimensionality reduction. IEEE Transactions on Image Processing, 27(8):4052–4065, 2018. doi: 10.1109/TIP.2018.2836312.
>
> [5] Bing Su et al. Heteroscedastic max-min distance analysis. In Proc. IEEE Conf. Comput. Vis. Pattern Recognit. (CVPR), pp. 4539–4547, 2015. doi: 10.1109/CVPR.2015.7299084.
>
> [6] Wei Bian and Dacheng Tao. Max-min distance analysis by using sequential SDP relaxation for dimension reduction. IEEE Transactions on Pattern Analysis and Machine Intelligence, 33(5):1037–1050, 2011. doi: 10.1109/TPAMI.2010.189.
>
>
> $\textbf{Final Remark:}$
>
> We again truly appreciate the time you have dedicated to evaluating our paper. We have applied your points for you to re-evaluate and hopefully gain your satisfaction to update your score. However, with all due respect, we believe some misunderstandings occurred that were not on our part (as explained in our responses), which likely affected your evaluation. Consequently, we feel that receiving a score of 2 with a confidence of 5 was unfair. We kindly ask you to re-examine our detailed responses and revised version, and we hope to achieve your complete satisfaction.

---

> ### Author Response · Authors · 2025-11-27
> **Invitation to Review Our Submitted Clarifications**
>
> Thank you again for the reviews. We wanted to briefly follow up because the points raised in the first round were important, and in our response we addressed all of them carefully with the goal of fully clarifying the contribution. Many of the clarifications directly resolve the earlier concerns, and we believe they present a much clearer and stronger picture of what the paper contributes. We would really appreciate it if you could take a moment to look at our reply. We posted our response early in the discussion period so there would be enough time for further back‑and‑forth if needed, but so far we have not received any follow‑up, and fewer than five days remain. We hope that the clarifications we provided can contribute to your assessment, including any consideration of adjusting the score if you feel it is appropriate. Any engagement at this stage would be very helpful.

---

> > ### Comment · Reviewer_GTno · 2025-11-27
> > **further clarification needed**
> >
> > Sorry I missed your response earlier (I thought I checked it but did not find any response, weird).
> >
> > Appreciate the detailed response. Would prefer if we could keep it shorter. There are a few things I couldn't follow and would appreciate a quick and short response:
> >
> > > If we add something that shows our approach is bounded from above, based on the monotone convergence theorem, the convergence in a finite number of steps is proven.
> >
> > After checking the appendix again (Line 773-787), I still do not follow how you can use monotonicity and boundedness to conclude convergence in finite time. Here is an example: the sequence 1/n is bounded from above and strictly decreases to 0, but we cannot conclude convergence in finite time. Did I miss anything else?
> >
> > > In other words, we did not do anything that would threaten optimality. In other words, we could prove globality for a non-convex problem without relaxation or using hyper-parameters.
> >
> > I do not follow this repeated claim on *relaxation-free* of the proposed algorithm. Eq (5), after an appropriate translation of S, is linear for the inner minimization problem but convex for the outer *maximization* problem. This is clearly a nonconvex problem? In Eq (7), you follow the MM framework to linearize the objective so that it becomes linear and hence convex for the outer problem too, from where you can apply the minimax theorem. But how is the linearization step "relaxation-free" and not threaten optimality? Does that mean applying the MM linearization trick can solve a nonconvex problem Eq (5)? Would appreciate some explanation here.

---

> ### Author Response · Authors · 2025-11-28
> **Response to Reviewer GTno (Second round)**
>
> We thank the reviewer for checking our response and for the follow-up questions. We appreciate the opportunity to clarify these two points briefly.
>
> $\textbf{1. On Finite Convergence vs. Asymptotic Convergence}$
>
> We now understand that your emphasis is on the word ``finite''. In our first-round response, we recognized this distinction and showed that while the objective is monotonically increasing and bounded above, this guarantees convergence to a limit, though not necessarily in a finite number of steps (as your $1/n$ example correctly illustrates).
>
> Consequently, in the revised manuscript submitted in the previous round, we already removed the word ``finite'' because it was not necessary for the proof. We instead describe the convergence over iterations $k$ (which implies asymptotic convergence). We appreciate you ensuring this mathematical precision.
>
> $\textbf{2. On ``Relaxation-Free'' and the MM Framework}$
>
> We emphasize the term "relaxation-free'' to distinguish our method from approaches that alter the problem definition (e.g., relaxing constraints to a convex hull), which create an optimality gap. Linearizing the objective via MM is not a ``relaxation'' in the traditional sense; it is an iterative optimization strategy.
>
> * Relaxation implies solving a simplified \textit{different} problem (often creating a gap).
> * MM involves constructing a surrogate function that locally touches the \textit{exact} objective function. By solving this iteratively, we reach a solution to the original problem.
>
> As detailed in Sun et al. [1] (specifically the subsection "Maximizing a Convex Function Over a Compact Set"), linearizing a convex objective function over a compact set and solving it iteratively is a rigorous method to find solutions for the original problem. Therefore, the linearization drives the algorithm toward a valid solution of the original non-convex problem without introducing the structural gaps associated with relaxation.
>
> We hope these clarifications satisfactorily address your concerns. We remain fully available and warmly welcome any further questions you may have, as we are committed to addressing all points to your complete satisfaction in hopes of meriting a positive re-evaluation of our work and a significant improvement in the score.
>
> [1] Y. Sun, P. Babu and D. P. Palomar, ``Majorization-Minimization Algorithms in Signal Processing, Communications, and Machine Learning,'' \textit{IEEE Transactions on Signal Processing}, vol. 65, no. 3, pp. 794--816, 2017.

---

### Official Review · Reviewer_s5YV · 2025-10-29

**Soundness:** 2
**Presentation:** 2
**Contribution:** 2
**Rating:** 2
**Confidence:** 4

**Summary:**

This paper studies discriminative dimensionality reduction, based on maximizing the minimum pairwise ratio of between-class to within-class scatter. The main technical development is on the optimization procedure of the objective defined in (2), which involves maximization over the linear projection and minimization over pairs of classes for finding the worst pair. The paper treats the problem as max-min fractional program and applies the generalize Dinkelbach’s algorithm.

**Strengths:**

The authors provide proofs for each statement.

**Weaknesses:**

1. I might be wrong and am willing to re-evaluate the paper if the authors can address my question:  for equation (4), after finding out the worse pair (i,j) for the inner minimization (e.g., by enumeration), cannot we solve the outer maximization problem, which is a generalized eigenvalue problem, with efficient numerical procedures? If that is the case, I find it hard to justify the convex relaxation on the constraint of $T$.

2. The datasets are tiny and are rarely used in modern ML papers. It is hard to evaluate the effectiveness and efficiency of methods based on these datasets.

**Questions:**

See weakness above. Also the particular approach seems too complicated to generalize the method to the deep learning framework.

---

> ### Author Response · Authors · 2025-11-15
> **Response to Reviewer s5YV (First round) - part 1**
>
> We thank the reviewer for the time and effort dedicated to assessing our work. We appreciate the opportunity to address the weaknesses and questions you raised. In what follows, we provide detailed justifications for our methodological choices, which we hope will clarify the novelty and rigor of our contribution.
>
> $\textbf{1. Why the Suggested Enumeration Approach is Fundamentally Flawed}$
>
> The reviewer's suggested approach—enumerating all pairs $(i,j)$ to find the worst case and then solving the outer maximization as a generalized eigenvalue problem—suffers from critical theoretical and computational limitations that render it unsuitable for solving the max-min problem as formulated in our paper.
>
> $\text{a) Circular Dependency - The Fatal Flaw:}$
>
>  The most critical issue is that the worst-case pair, $(i,j)^*$, $\textbf{is not independent of the projection matrix $\mathbf{T}$}$; it is, in fact, a function of $\mathbf{T}$. This creates an unresolvable circular dependency:
>
> * To identify the worst pair $(i,j)^\*$, one must already know the optimal projection $\mathbf{T}^\*$.
>
> * To compute the optimal projection $\mathbf{T}^\*$, one must first know which pair $(i,j)^\*$ is the worst.
>
> If we fix a pair $(i,j)$ and solve the resulting generalized eigenvalue problem to obtain a projection $\mathbf{T}{(i,j)}^\*$, there is $\textbf{absolutely no guarantee}$—neither theoretical nor empirical—that this same pair $(i,j)$ remains the worst case for that specific solution $\mathbf{T}{(i,j)}^*$. Solving for different fixed pairs will generally yield different optimal projections, each making a different pair the new worst case. A simple enumeration or alternating heuristic $\textbf{cannot}$ resolve this fundamental interdependency.
>
> $\text{b) No Guarantee of Global Optimality:}$
>
>  An iterative alternating procedure (e.g., fix $\mathbf{T}$, find the worst pair; then fix the pair, solve for $\mathbf{T}$; repeat) is essentially a heuristic that:
> * $\text{Will almost certainly converge to a suboptimal local solution}$ rather than the true global optimum of the original max-min problem.
> * $\text{Lacks any theoretical convergence guarantees}$ to the correct solution.
> * $\text{Does not satisfy the Karush-Kuhn-Tucker (KKT) optimality conditions}$ of the original max-min problem.
>
> $\textbf{2. On Dataset Choice and Scalability}$
>
> Regarding the use of three comparison datasets and the question of scalability, we offer the following clarification.
>
> The datasets used in our experiments are established high-dimensional benchmarks within the discriminative dimensionality reduction literature [1-5]. But even if you dismiss this evidence, and we accept with high confidence that those datasets are not extremely high-dimensional, a comprehensive review of post-2015 literature confirms that the field no longer struggles with raw high-dimensionality for this class of problems. This is because $\text{all}$ $\text{modern competitors}$ $\text{first apply}$ $\text{Principal Component Analysis (PCA)}$ $\text{as a standard preprocessing step.}$
>
> For example, they mention that if they want to deal with a dataset with a high dimensionality of 1024, they apply PCA to reduce the dimension to below 50 while keeping 98% of the energy of the dataset. We also noted our use of PCA in the original manuscript. This step is a practical necessity; most competing methods rely on Semidefinite Programming (SDP) solvers (e.g., CVX, CVXPY), which would face prohibitive memory constraints and crash if applied to the raw high-dimensional covariance matrices.
>
> Consequently, the primary research focus in this domain is no longer on handling raw high-dimensionality but on improving $\text{computational efficiency,}$ $\text{memory usage,}$ $\text{and classification accuracy}$ $\text{in the post-PCA reduced space}$—which is precisely the contribution of our paper. Because of the importance of this issue, we have added a related explanation to the introduction section of the revised version of the paper, clarifying that this problem has already been solved.

---

> ### Author Response · Authors · 2025-11-15
> **Response to Reviewer s5YV (First round) - part 2**
>
> $\textbf{Answering your questions}$
>
> $\text{On Generalization to Deep Learning Frameworks:}$
>
> Regarding your question about the complexity of generalizing the method to a deep learning framework, we have some strong answers that we hope will convince you.
>
> a) First of all, we have provided a solver that is relaxation-free, hyperparameter-free, and for which we provide proof of convergence to a high-quality solution. With all due respect, we do not agree with you about it being complicated because, in contrast, we have provided an approach that is training-free (a common requirement for deep learning-based methods).
>
> b) We have done a full study of the literature. With high confidence, we can assure you that there is no neural network-based paper dealing with worst-case class separation (a max-min cost) to compare against. The reason for this may be that defining a max-min approach in class separation requires maximizing over a continuous variable while having an inner discrete variable. However, as another advantage of our approach, we have proposed a method that, by applying the extended Dinkelbach approach and tricks from the MM framework, has the ability to interchange max-min to min-max. And, because the inner max has a closed-form solution, the problem is converted to a min optimization. While this min problem can be solved theoretically as we have done, it can also be handled by a neural network, which shows the advantage of our approach. In other words, we can contend that our paper is the first to propose an approach that can create a bridge from the theoretical solving of max-min problems to deep learning-based methods.
>
> $\textbf{Final Remarks}$
>
> We sincerely appreciate the time you have taken to review our work and are grateful for your willingness to re-evaluate it. After extensive research in this domain, we have developed a novel method for solving this max-min fractional quadratic problem that is simultaneously: $\textbf{1) exact (relaxation-free)}$, $\textbf{2) hyperparameter-free}$, $\textbf{3) training-free}$, $\textbf{4) computationally efficient}$, and $\textbf{5) highly accurate.}$ With high confidence, we assert that $\textbf{no published work achieves all of these properties concurrently.}$
>
> Given that our primary methodological contribution is theoretically sound and the suggested alternative is fundamentally flawed—a point you kindly acknowledged may have been an oversight—we were surprised by the recommendation for rejection based on the secondary concerns, which we believe have now been fully addressed. We hope our detailed explanations have clarified the significance of our work and its contributions. A rejection can have a profound impact on future academic paths, and we therefore kindly and respectfully ask you to re-evaluate our manuscript, in the hope that our score can be revised to reflect the strengths of this research.

---

> ### Author Response · Authors · 2025-11-27
> **Invitation to Review Our Submitted Clarifications**
>
> Thank you again for the reviews. We wanted to briefly follow up because the points raised in the first round were important, and in our response we addressed all of them carefully with the goal of fully clarifying the contribution. Many of the clarifications directly resolve the earlier concerns, and we believe they present a much clearer and stronger picture of what the paper contributes. We would really appreciate it if you could take a moment to look at our reply. We posted our response early in the discussion period so there would be enough time for further back‑and‑forth if needed, but so far we have not received any follow‑up, and fewer than five days remain. We hope that the clarifications we provided can contribute to your assessment, including any consideration of adjusting the score if you feel it is appropriate. Any engagement at this stage would be very helpful.

---

> ### Comment · Reviewer_s5YV · 2025-11-27
>
> 1. Frist of all, did the authors show comparison with the alternating optimization approach, where one alternate over two steps: finding the worse pair, and  solving generalized eigenvalue problem?
>
> 2. I am reading the proofs and I am trying to understand global convergence. Now, for line 724, I do not easily see how you go from $\min_{i,j} \frac{f_{ij}(T)}{g_{ij}(T)} \le \lambda^*$
> to the next step that
>
> $f_{ij} (T) - \lambda^* g_{ij}(T) \le 0$ for all pairs $(i,j)$?

---

> ### Author Response · Authors · 2025-11-28
> **Response to Reviewer s5YV (Second round)**
>
> 1) To investigate the feasibility of the suggested alternating optimization approach (solving the generalized eigenvalue problem for the “worst” pair iteratively), we implemented a simulation using synthetic data. As illustrated in the figure provided in the supplementary material, this greedy strategy fails to converge.
>
>    Aligned with the explanation in our first-round response, the algorithm does not converge to a solution and enters a continuous limit cycle. Specifically, strictly optimizing the projection matrix $\mathbf{T}$ for the current worst pair (e.g., pair 3) inevitably degrades the ratio of another pair (e.g., pair 6). We have provided both the generated plot and the corresponding MATLAB code in the supplementary material to facilitate the reproduction of this result."
>
> 2) You are correct, and we appreciate you pointing this out. We kindly ask the reviewer to recheck the revised proof provided in the updated manuscript. The corrected derivation relies on the minimum of the set.
>
> We hope these clarifications satisfactorily address your concerns. We remain fully available and warmly welcome any further questions you may have, as we are committed to addressing all points to your complete satisfaction in hopes of meriting a positive re-evaluation of our work and a significant improvement in the score.

---

### Official Review · Reviewer_heDT · 2025-10-30

**Soundness:** 3
**Presentation:** 3
**Contribution:** 3
**Rating:** 6
**Confidence:** 3

**Summary:**

This paper proposes a novel discriminative dimensionality reduction method, GDMM-QF, based on maximizing the minimum pairwise ratio of between-class to within-class scatter (PWCRA). Specifically, in the outer loop, a generalized Dinkelbach-type procedure to transform the challenging maxmin fractional objective into an equivalent max-min subtractive problem. In the inner loop, the Minorization-Maximization (MM) framework is adopted to construct a tight, convex surrogate that lower-bounds the true objective, resulting in a semidefinite program (SDP) at each inner step. Finally, extensive experiments on multiple datasets demonstrate the superiority of the proposed method in learning discriminative projections.

**Strengths:**

1. The paper introducing the Pairwise Worst-Case Ratio Analysis (PWCRA) objective. This is a improvement over existing worst-case methods because it uses a pairwise-adaptive within-class scatter matrix, making it more robust to heteroscedastic data where class variances differ significantly.
2.  The theoretical foundation is strong. The paper provides rigorous proofs for all key claims, including the equivalence of the original problem to a root-finding problem (Theorem 1), the convergence of the outer loop to the global optimum (Theorem 3), and the validity of the convex relaxation (Lemma 1).
3. The proposed GDMM-QF algorithm addresses a fundamental challenge in discriminant analysis: ensuring robust separation even for the most difficult class pairs.

**Weaknesses:**

1.	The author only provided the mean and standard deviation (based on 20 repetitions) but did not perform any significance tests (such as paired t-test) to demonstrate whether the improvement was statistically significant.
2.	The paper mentions that the complexity of the QP method is O(C⁶ + d³). However, the paper does not validate this on larger datasets. Could you provide such validation?

**Questions:**

1. The results in Table 1 show impressive performance gains. Could the authors perform a statistical significance test (such as paired t-test) to confirm that the differences in error rates between GDMM-QF and the next best method are statistically significant?
2. It would be better to have validation with a larger dataset.

---

> ### Author Response · Authors · 2025-11-15
> **Response to Reviewer heDT (First round) - part 1**
>
> Dear Reviewer Hdet,
>
> We are very thankful for the time and endeavor you invested in examining our paper. We truly appreciate that you found the following significant strengths:
>
> * The introduction of the Pairwise Worst-Case Ratio Analysis (PWCRA) objective, which is an improvement over existing worst-case methods because it uses a pairwise-adaptive within-class scatter matrix, making it more robust to heteroscedastic data where class variances differ significantly.
>
> * The strong theoretical foundation, with rigorous proofs for all key claims, including the equivalence of the original problem to a root-finding problem (Theorem 1), the convergence of the outer loop to the global optimum (Theorem 3), and the validity of the convex relaxation (Lemma 1).
>
> * The fact that the proposed GDMM-QF algorithm addresses a fundamental challenge in discriminant analysis: ensuring robust separation even for the most difficult class pairs.
>
> * Regarding the weaknesses you mentioned, we have provided our responses below. To be honest, our goal is to fully convince you and earn the highest possible score.
>
> $\text{1. On the Lack of Significance Tests:}$
>
> Regarding your concern, as you can see in the table 1 from the paper, we strived to obtain the mean and standard deviation of the error rate. It is worth mentioning that in contrast to papers like [1] which only focus on 1-NN, we wanted to have a strong evaluation, so we chose three metrics (1-NN, NM, and QDA) to evaluate performance. Regarding the t-test, because of the page limitation, we did not include one, but we followed the established protocol in the literature. The use of 20 repetitions is exactly the value used in [1-5]; to assure you of this, we have taken their direct quotes as below:
>
> [1]: “We randomly select 100 samples in each class for training and the rest of samples for testing. All the results are the average value by conducting each group of experiment 20 times repeatedly.”
>
> [2]: “For the high-dimensional Digit dataset, we followed Wang et al. (2024), performing 20 independent experimental runs where 50% of samples were randomly selected for training and the remainder for testing in each run. PCA preprocessing was applied to all datasets following Wang et al. (2024); Su et al. (2015), preserving 98% of the variance.”
>
> [3]: “The experiments are repeated 10 times, and the best average recognition rates are recorded.”
>
> [4]: “we randomly split the whole database to generate the training and test sets 10 times, and take the average error rates and standard deviations as evaluation indicators.”
>
> [5]: “we randomly split the whole database to generate the training and test sets 10 times, and take the average error rates and standard deviations as evaluation indicators.”
>
> $\text{2. On Validation with Larger Datasets:}$
>
> First of all, you mentioned using high-dimensional data. In the class separation application, after 2015 the concern has not been on high-dimensional data anymore because papers apply PCA to reduce the dimension to below 50 and then apply their proposed method. In other words, after 2015, the focus has been on finding approaches that have 1) maximum accuracy, 2) maximum computational efficiency, and 3) minimal memory usage for the post-PCA dimensions (because applying PCA on high-dimensional datasets reduces them to below 50 dimensions). Again, we kindly ask you to read papers [1-5] (which are baselines in class separation), and you will notice that they have all applied PCA. Besides, we mention that if they did not do this, because most of them are SDP-based, they could not apply CVX or CVXPY and their systems would absolutely crash. Because of the importance of this issue, we have added a related explanation to the introduction section of the revised version of the paper, clarifying that this problem has already been solved.
>
> In our work, our approach is 1) relaxation-free and 2) hyperparameter-free. Furthermore, 3) by using max-min on ratio analysis, we not only strive to separate the worst-case (extremely overlapped) classes, but the normalization also serves to concentrate the points within each class and avoid high inner-class variance. With our approach, we could achieve better accuracy compared to competitors while having reliable computational efficiency. We kindly ask the reviewer to reexamine this point.

---

> ### Author Response · Authors · 2025-11-15
> **Response to Reviewer heDT (First round) - part 2**
>
> $\textbf{final remarks}$
>
> Dear respectful reviewer, we truly appreciate the time you dedicated to our paper. We hope that with the above explanations and evidence we have provided, we have been able to convince you completely. We want to earn your highest score, so if you think there are still some issues that must be clarified, we are open to discussing them further.
>
> [1] Zheng Wang et al. Worst-case discriminative feature learning via max-min ratio analysis. IEEE Trans. Pattern Anal. Mach. Intell., 46(1):641–658, 2024. doi: 10.1109/TPAMI.2023.3323453.
>
> [2] Mohammad Mahdi Omati, Prabhu babu, Petre Stoica, and Arash Amini. A max-min approach to the worst-case class separation problem. Transactions on Machine Learning Research, 2025. ISSN 2835-8856. URL https://openreview.net/forum?id=EEmwBd4tfZ.
>
> [3] Feiping Nie, Zheng Wang, Rong Wang, Zhen Wang, and Xuelong Li. Towards robust discriminative projections learning via non-greedy ℓ2,1-norm minmax. IEEE Trans. Pattern Anal. Mach. Intell., 43(6):2086–2100, 2021b.
>
> [4] Bing Su, Xiaoqing Ding, Changsong Liu, and Ying Wu. Heteroscedastic max–min distance analysis for dimensionality reduction. IEEE Transactions on Image Processing, 27(8):4052–4065, 2018. doi: 10.1109/TIP.2018.2836312.
>
> [5] Bing Su et al. Heteroscedastic max-min distance analysis. In Proc. IEEE Conf. Comput. Vis. Pattern Recognit. (CVPR), pp. 4539–4547, 2015. doi: 10.1109/CVPR.2015.7299084.

---

> ### Author Response · Authors · 2025-11-27
> **Invitation to Review Our Submitted Clarifications**
>
> Thank you again for the reviews. We wanted to briefly follow up because the points raised in the first round were important, and in our response we addressed all of them carefully with the goal of fully clarifying the contribution. Many of the clarifications directly resolve the earlier concerns, and we believe they present a much clearer and stronger picture of what the paper contributes. We would really appreciate it if you could take a moment to look at our reply. We posted our response early in the discussion period so there would be enough time for further back‑and‑forth if needed, but so far we have not received any follow‑up, and fewer than five days remain. We hope that the clarifications we provided can contribute to your assessment, including any consideration of adjusting the score if you feel it is appropriate. Any engagement at this stage would be very helpful.

---

### Official Review · Reviewer_XeLp · 2025-11-02

**Soundness:** 2
**Presentation:** 2
**Contribution:** 2
**Rating:** 4
**Confidence:** 3

**Summary:**

This paper presents GDMM-QF, a novel discriminative dimensionality reduction method that maximizes the minimum pairwise ratio of between-class to within-class scatter, thereby enhancing class separability through adaptive control of class-pair variances. GDMM-QF addresses the resulting non-convex max-min fractional programming problem. GDMM-QF is hyperparameter-free, computationally efficient, and guarantees convergence. Extensive experiments on benchmark datasets demonstrate its effectiveness in learning highly discriminative low-dimensional representations, consistently outperforming existing state-of-the-art methods in classification accuracy.

**Strengths:**

1.	The proposed Dinkelbach–Minorization–Maximization framework is novel and contributes a fresh perspective to the field of Artificial Intelligence.
2.	The paper provides a thorough and rigorous mathematical analysis of the GDMM-QF method.
3.	The method consistently outperforms baselines (LDA, MMDA, MMRA, etc.) across multiple benchmark datasets.
4.	The algorithm’s parameter-free design is a notable practical advantage, reducing the need for extensive hyperparameter tuning and enhancing reproducibility in real-world applications.

**Weaknesses:**

1.	The experiments are conducted on only three datasets. Evaluating the method on larger-scale or more complex datasets would strengthen the claims regarding scalability and generalizability.
2.	Although the paper claims efficiency improvements over SDP-based methods, the empirical section lacks detailed runtime comparisons or scalability analyses as the dataset dimensionality increases.
3.	The paper does not analyze the impact of key components (e.g., pairwise normalization, MM approximation) individually. Including ablation studies could help clarify the source of the observed performance gains.
4.	The paper lacks visual illustrations such as framework or pipeline diagrams and primarily focuses on theoretical analysis. Incorporating such visualizations would improve clarity and accessibility.

**Questions:**

1.	It might be helpful if the authors could include empirical comparisons of runtime and memory usage on higher-dimensional or large-scale datasets to further support the claimed efficiency and scalability.
2.	The paper introduces a new optimization framework, but some parts of the motivation and intuition behind the method could be clearer. Could the authors provide more intuition or illustrative examples to help readers understand why this approach works better than existing ones?
3.	It appears that the proposed method does not rely on neural networks. So it could be useful to discuss whether there are related neural network–based approaches addressing the same problem, and how the proposed method compares with them in terms of advantages and limitations.
4.	To ensure reproducibility, will the authors release the source code and experimental setup? If not, additional implementation details—such as initialization strategy, stopping criteria, and computational resources—would be appreciated.

---

> ### Author Response · Authors · 2025-11-15
> **Response to Reviewer XeLp (First round) - part 1**
>
> Dear Reviewer XeLp,
>
> We are sincerely grateful for the time and insight you dedicated to reviewing our work. We were particularly encouraged that you found value in several key aspects of our research:
>
> * The innovative nature of our Dinkelbach–Minorization–Maximization framework.
> * The thoroughness and rigor of the mathematical proofs for the GDMM-QF method.
> * The consistent and superior performance of our method on benchmark datasets.
> * The parameter-free design, which enhances its reproducibility and practical utility.
>
> Regarding the weaknesses, we have strived to address your concerns one by one. We are confident that with the answers below, we can convince you completely.
>
> $\textbf{Response to Weaknesses}$
>
> $\text{1. On the Number of Datasets and Scalability:}$
>
> In response to this, it is worth mentioning that we strove to work on datasets (three datasets for comparison and one extra dataset for visualization) that have been introduced in the literature as high-dimensional datasets. We kindly ask the reviewer to see [1-5], which are a few of the papers that use these datasets. But even if you think this is not good evidence, and we accept that those datasets are not extremely high-dimensional, we offer the following point. Dear reviewer, based on our complete literature review of all related papers from 1948 to the present for the class separation application in linear discriminant analysis, since 2015, there is no longer a challenge related to working on high-dimensional datasets. This is because all baseline methods first apply PCA to lower the dimensions and then apply their proposed method. We kindly ask you to check [1-5] again, which explicitly mention that for high-dimensional data they apply this preprocessing. For example, they mention that if they want to deal with a dataset with a high dimensionality of 1024, they apply PCA to reduce the dimension to below 50 while keeping 98% of the energy of the dataset. It is worth mentioning that we had mentioned using PCA in the original paper as below:
>
> " As a preprocessing step, following the protocol of Omati et al. (2025); Wang et al. (2024); Su et al. (2015), we apply PCA to project all feature vectors down to 50 dimensions, thereby retaining over 98% of the total variance. We repeat this entire process 20 times with independent random splits and report the mean accuracy and its standard deviation."
>
> Besides, we want to provide another piece of evidence. As mentioned in the paper several times, most competitors rely on relaxation + SDP. If they did not apply PCA, how could they handle solving their extremely high-dimensional matrices in CVX? Without applying PCA, CVX and CVXPY would crash due to the memory they would occupy. So, based on what is mentioned in the baseline papers, after 2015, the main concern is not dealing with extremely high-dimensional data; rather, papers focus on how they can improve computational and memory efficiency while maintaining an extremely high level of accuracy. Because of the importance of this issue, we have added a related explanation to the introduction section of the revised version of the paper, clarifying that this problem has already been solved.
>
> $\text{2. On Runtime Comparisons:}$
>
> Regarding your concern that the empirical section lacks detailed runtime comparisons or scalability analyses, with all due respect, we think an oversight may have occurred. As you know, ICLR has a page limitation. Therefore, we had to place additional experimental results in the supplementary materials (which are located after the references). If you check that section, you will notice we had provided a comparison across four dimensions for the three datasets to show the efficiency of our proposed method.

---

> ### Author Response · Authors · 2025-11-15
> **Response to Reviewer XeLp (First round) - part 2**
>
> $\text{3. On the Analysis of Key Components (Ablation Study):}$
>
> Regarding the point about not analyzing the impact of key components (e.g., pairwise normalization, MM approximation), we kindly ask the reviewer to recheck our original paper. In the original paper, we mention that the paper [1] strives to solve WCRA, but their solver has a problem because they apply relaxation + SDP, which moves the solution far from a high-quality one (this exact sentence is also mentioned in [2]). Besides, another advantage of our proposed method relative to theirs was that it is hyperparameter-free, while they need two hyperparameters that must be chosen by searching within an interval. Therefore, our comparison with [1] serves as an ablation of using the MM approach versus not using it.
>
> Related to the point you made about an ablation on normalization, as mentioned in the paper, [2,3,4] strove to deal with a non-normalized cost (which in this application is called a distance- or quadratic-based cost). Of these, [3,4] strove to solve the problem with relaxation + SDP and suffered from drawbacks like moving far away from a high-quality solution and being hyperparameter-based. Method [2] could solve their distance-based (or non-normalized cost) problem with MM. We kindly ask the reviewer to recheck the original paper because we had comparisons covering all the situations that were of concern to you. For your convenience, in the revised version of the paper, we have strived to make these points clearer.
>
> $\text{4. On the Lack of Visual Models:}$
>
> Dear reviewer, as you mentioned, our proposed method is fully theoretical. We agree with you that if our proposed method were deep learning-based, a visual model could have made it more powerful. However, because our paper is theory-based, like all papers that deal with purely theoretical contributions, we have written pseudocode which was included in the supplementary file. Because of the point mentioned in your weakness #2, you probably did not check the supplementary material to see that. It is worth mentioning that we did not intend to include the pseudocode in that section; however, due to the page limitation, we had to do so. For the camera‑ready version, since we will be granted one additional page, we plan to move it back to the main paper.

---

> ### Author Response · Authors · 2025-11-15
> **Response to Reviewer XeLp (First round) - part 3**
>
> $\textbf{Answering your Questions:}$
>
> 1. $\text{Runtime on high-dimensional data}$: As mentioned in the response to the weaknesses, we kindly ask you to reconsider our points. The standard practice in the field is to first apply PCA to high-dimensional data. This means the primary challenge is no longer about raw dimensionality but about efficiency and accuracy in the post-PCA space, which our paper already addresses.
>
> 2. $\text{Intuition behind the method}$: Related to this question, as mentioned in the paper, competitors strive to either deal with a distance-based cost (other names: quadratic-based or non-normalized) [2,4,5] or a ratio-based cost (fractional quadratic-based or normalized) [1,3]. Besides, some of them were relaxation-based [3,4,5] and some were MM-based (which, as mentioned, were for a distance-based cost) [2]. As mentioned in the response to the weaknesses, we have done a full ablation on this. Besides, regarding the effect that normalization, the kind of solver, etc., can have, we have provided a visual comparison in Figure 1.
>
> 3. $\text{Comparison with Neural Networks}$: Yes, as you noticed, our paper is not neural-network-based. We have done a full study of the literature. The papers that deal with class separation focus more on theory. There are some papers that attempt to deal with linear discriminant analysis in the deep learning space, but their results were only competitive with LDA (which is a completely theoretical approach), and LDA has advantages because it does not require training (as is common in neural networks). With high confidence, we can ensure you that we have done a complete review of the literature, and there is no neural networks-based paper that has tried to deal with worst-case class separation (a max-min cost) that we could compare with. The reason that neural network-based methods do not deal with worst-case separation may come from the fact that defining a max-min approach in class separation requires performing a max over a continuous variable while having an inner discrete variable. However, as another advantage of our approach, we could propose a method that, by applying the extended Dinkelbach method and tricks from MM, has the ability to interchange max-min to min-max. And because the inner max has a closed-form solution, the problem converts to a min optimization. While this min problem can be solved theoretically, it could also be handled with a neural network, which shows an advantage of our approach.
>
> 4. $\text{Code and Reproducibility}$: Absolutely. Upon acceptance of the paper, we guarantee that we will share the code on GitHub. Related to the points you mentioned, we have added the below information to the revised version:
>
>    “All experiments were conducted in MATLAB R2022b on a dual‑socket Intel Xeon E5‑2695 v3 workstation equipped with 2 × 14 cores (56 threads total), operating at a base frequency of 2.3 GHz (up to 3.3 GHz turbo) and featuring 70 MiB of L3 cache."

---

> ### Author Response · Authors · 2025-11-15
> **Response to Reviewer XeLp (First round) - part 4**
>
> $\textbf{Final Remarks}$
>
> Dear reviewer, first of all, we very much appreciate the time you dedicated to the examination of our paper. We fully understand that this work requires significant time and accuracy to examine, and we are sure you have strived to do that. That is really valuable for us. But, with all due respect, we think some misunderstandings may have occurred that were not on our side, which probably had an effect on your score evaluation, and we kindly ask you to reexamine our paper. We have mentioned them below:
>
> 1) You mentioned the use of high-dimensional data, while we strove to use datasets that are introduced as high-dimensional in the literature (e.g., COIL-20 and Yale). In the class separation application after 2015, the primary concern has no longer been high-dimensional data because even for the most extremely high-dimensional datasets, PCA is used as a preprocessing step to create a lower-dimensional representation before their proposed method is applied. In other words, after 2015, the focus has been on finding approaches that have 1) maximum accuracy, 2) maximum computational efficiency, and 3) minimal memory usage for the post-PCA dimensions (because PCA on high-dimensional datasets reduces them to below 50 dimensions). Again, we kindly ask you to read papers [1-5] (which are baselines in class separation), and you will notice they have all applied PCA as preprocessing. Besides, we reiterate that if they did not do this, because most of them are SDP-based, they could not use CVX or CVXPY and would certainly experience system crashes.
>
> 2) You mentioned the lack of runtime comparisons, while we already had them in the appendix after the references, which suggests this section may have been overlooked.
>
> 3) Regarding the ablation study, as explained in the response to the weaknesses, we had already examined diverse situations (with or without normalization, with or without relaxation, using MM or not using it).
>
> As a result, while we truly appreciate your valuable points and hope we have convinced you with our responses, we believe, with all due respect, that some issues you mentioned did not originate from our work, which had an effect on the score you gave us. If you think our explanation has convinced you, it would be extremely appreciated if you would re-evaluate and update your score. Furthermore, if you believe any points still require more discussion, we are entirely open to it, as our primary goal is to address all of your concerns to your complete satisfaction. Had we not spent a long time working on this paper and the literature, we could not guarantee our paper with such high confidence.
>
> [1] Zheng Wang et al. Worst-case discriminative feature learning via max-min ratio analysis. IEEE
> Trans. Pattern Anal. Mach. Intell., 46(1):641–658, 2024. doi: 10.1109/TPAMI.2023.3323453.
>
> [2] Mohammad Mahdi Omati, Prabhu babu, Petre Stoica, and Arash Amini. A max-min approach to the
> worst-case class separation problem. Transactions on Machine Learning Research, 2025. ISSN
> 2835-8856. URL https://openreview.net/forum?id=EEmwBd4tfZ.
>
> [3] Feiping Nie, Zheng Wang, Rong Wang, Zhen Wang, and Xuelong Li. Towards robust discriminative
> projections learning via non-greedy ℓ2,1-norm minmax. IEEE Trans. Pattern Anal. Mach. Intell.,
> 43(6):2086–2100, 2021b.
>
> [4] Bing Su, Xiaoqing Ding, Changsong Liu, and Ying Wu. Heteroscedastic max–min distance analysis
> for dimensionality reduction. IEEE Transactions on Image Processing, 27(8):4052–4065, 2018.
> doi: 10.1109/TIP.2018.2836312.
>
> [5] Bing Su et al. Heteroscedastic max-min distance analysis. In Proc. IEEE Conf. Comput. Vis. Pattern
> Recognit. (CVPR), pp. 4539–4547, 2015. doi: 10.1109/CVPR.2015.7299084.

---

> ### Author Response · Authors · 2025-11-27
> **Invitation to Review Our Submitted Clarifications**
>
> Thank you again for the reviews. We wanted to briefly follow up because the points raised in the first round were important, and in our response we addressed all of them carefully with the goal of fully clarifying the contribution. Many of the clarifications directly resolve the earlier concerns, and we believe they present a much clearer and stronger picture of what the paper contributes. We would really appreciate it if you could take a moment to look at our reply. We posted our response early in the discussion period so there would be enough time for further back‑and‑forth if needed, but so far we have not received any follow‑up, and fewer than five days remain. We hope that the clarifications we provided can contribute to your assessment, including any consideration of adjusting the score if you feel it is appropriate. Any engagement at this stage would be very helpful.

---

### Note · Authors · 2026-02-25

I have read and agree with the venue's withdrawal policy on behalf of myself and my co-authors.

---

### Meta-Review · Area_Chair_Uzb6 · 2026-01-03

**Summary:**

Reviewer XeLp recognized the novelty of the proposed framework and this paper’s rigorous mathematical analysis. But the reviewer has the following major concerns.
1. Only evaluating on three datasets is insufficient.
2. The running time evaluation is lacking in this paper.
3. This paper lacks a detailed ablation study.

The author states that the running time comparison is provided in the appendix. And the required ablation study has been performed in the paper. The authors did not offer an evaluation on more datasets.

Reviewer heDT recognizes the novelty and solid theoretical foundation of the proposed method. The Reviewer suggests that the author provide a statistical significance test and evaluate the proposed method on a large dataset. But the authors did not follow the reviewer’s suggestions.

Reviewer s5YV also points out that the proposed method was only evaluated on tiny datasets.

Reviewer GTno also questions the small scale of the datasets. The Reviewer also thinks the Theorem and Lemma statements have gaps. The authors admit that the concerns from the Reviewer on the rigor of the Theorems, and will fix them in the revised version.

**Reviewer Concerns:**

The concerns about running time and the ablation study are well solved. But the problems with evaluation on larger-scale datasets are not solved by the authors.

**Reviewer Scores:**

I do not think the reviewers will change their ratings.

---

### Decision · Program_Chairs · 2026-01-26

Reject